# Local environment in biomolecular condensates modulates enzymatic activity across length scales

Marcos Gil-Garcia [1], Ana I. Benítez-Mateos[2], Marcell Papp[1], Florence Stoffel[1], Chiara Morelli [1], Karl Normak[1], Katarzyna Makasewicz[1], Lenka Faltova[1], Francesca Paradisi [2] & Paolo Arosio [1] ✉

The mechanisms that underlie the regulation of enzymatic reactions by biomolecular condensates and how they scale with compartment size remain poorly understood. Here we use intrinsically disordered domains as building blocks to generate programmable enzymatic condensates of NADH-oxidase (NOX) with different sizes spanning from nanometers to microns. These disordered domains, derived from three distinct RNA-binding proteins, each possessing different net charge, result in the formation of condensates characterized by a comparable high local concentration of the enzyme yet within distinct environments. We show that only condensates with the highest recruitment of substrate and cofactor exhibit an increase in enzymatic activity. Notably, we observe an enhancement in enzymatic rate across a wide range of condensate sizes, from nanometers to microns, indicating that emergent properties of condensates can arise within assemblies as small as nanometers. Furthermore, we show a larger rate enhancement in smaller condensates. Our findings demonstrate the ability of condensates to modulate enzymatic reactions by creating distinct effective solvent environments compared to the surrounding solution, with implications for the design of protein-based heterogeneous biocatalysts.

The ability of cells to compartmentalize reactions in space and time is pivotal for the coordination of cellular metabolism. In addition to membrane-bound organelles, it has emerged that cells can create membraneless-compartments through the de-mixing of proteins and nucleic acids from their surrounding milieu[1–4].

These biomolecular condensates consist of different scaffold biomolecules and are mediated by multivalent, weak interactions that can be encoded through various mechanisms, including multiple folded domains, intrinsically disordered regions, and protein-nucleic acid interactions[5]. In many cases, a class of intrinsically disordered regions known as low-complexity domains (LCDs) has been shown to drive or modulate phase separation both in vivo and under diverse solution conditions in vitro[6–9].

The emergent properties of biomolecular condensates (such as the recruitment of client molecules, viscoelasticity, and surface tension) can be finely adjusted through the design of their scaffold proteins, including LCDs. This inherent versatility makes biomolecular condensates based on LCDs attractive protein-based materials for various biomedical and biotechnological applications. They can be incorporated into the array of available tool-boxes (which include, for instance, elastin-like and other thermoresponsive peptides[10–12], amyloids[13–16], nanocages[17,18], and coiled-coils[19,20]) for creating protein-based supramolecular architectures with tailored functionalities.

Condensates can find applications as enzymatic microreactors in heterogeneous biocatalysis. Numerous enzyme immobilization

---

[1]Department of Chemistry and Applied Biosciences, Institute for Chemical and Bioengineering, ETH Zurich, Zurich, Switzerland. [2]Department of Chemistry, Biochemistry and Pharmaceutical Sciences, University of Bern, Bern, Switzerland. ✉e-mail: paolo.arosio@chem.ethz.ch

methods depend on the covalent conjugation of enzymes to inorganic scaffolds, a process that may compromise their activity[21,22]. In contrast, condensates offer a promising alternative since enzymes are situated within a native-like environment, and the quaternary assemblies can be genetically encoded, thus avoiding the need for harsh chemical conjugation steps[23].

Developing cell-mimicking condensates in test tubes can also serve as a valuable tool for unraveling mechanisms involved in modulating biochemical reactions within cells. Condensation can affect enzymatic reactions through various mechanisms, which go beyond mere increases in the local concentration of enzymes and substrates[24-26]. Moreover, condensates within cells can populate not only the microscale but also the nanoscale[27], potentially being smaller than the micron-sized droplets observed in test tubes. It remains unclear if and how the effect of condensation on enzymatic reactions changes across different length scales.

Examples of enzymes found in biological condensates include DEAD-box helicases, which modulate the properties of P bodies[7]. These RNA-binding proteins contain LCDs that modulate phase separation[7]. Although the effect of LCDs in helicases may involve interactions with the globular domains[28], it has been demonstrated

that intrinsically disordered regions can be used in a modular way as building blocks to induce phase separation of soluble enzymes and generate synthetic enzymatic condensates both in vitro and within cells[29-35].

In this work, we use different LCDs derived from DEAD-box helicases to generate condensates characterized by distinct microenvironments and spanning a broad size range from clusters of hundreds of nanometers to micron-sized droplets. Our aim is to investigate the effect of microenvironment and size on enzymatic activity. Specifically, we selected LCDs with different net charges to modulate the recruitment of the negatively-charged substrate and cofactor, and to investigate their effects on the activity of the model enzyme NADH oxidase (NOX) from *Thermus thermophilus*[36]. NOX catalyzes the oxidation of NADH into NAD$^+$ in the presence of the cofactor FAD, utilizing oxygen and generating hydrogen peroxide (H$_2$O$_2$) (Fig. 1A)[37]. NOX enzymes are crucial for NAD$^+$ regeneration systems involved in NAD-dependent dehydrogenase-catalyzed reactions, such as the oxidation of alcohols via alcohol dehydrogenases[38]. With an isoelectric point of 8.86, the enzyme has a propensity to assemble in dimers, therefore further promoting multivalent interactions that drive phase separation.

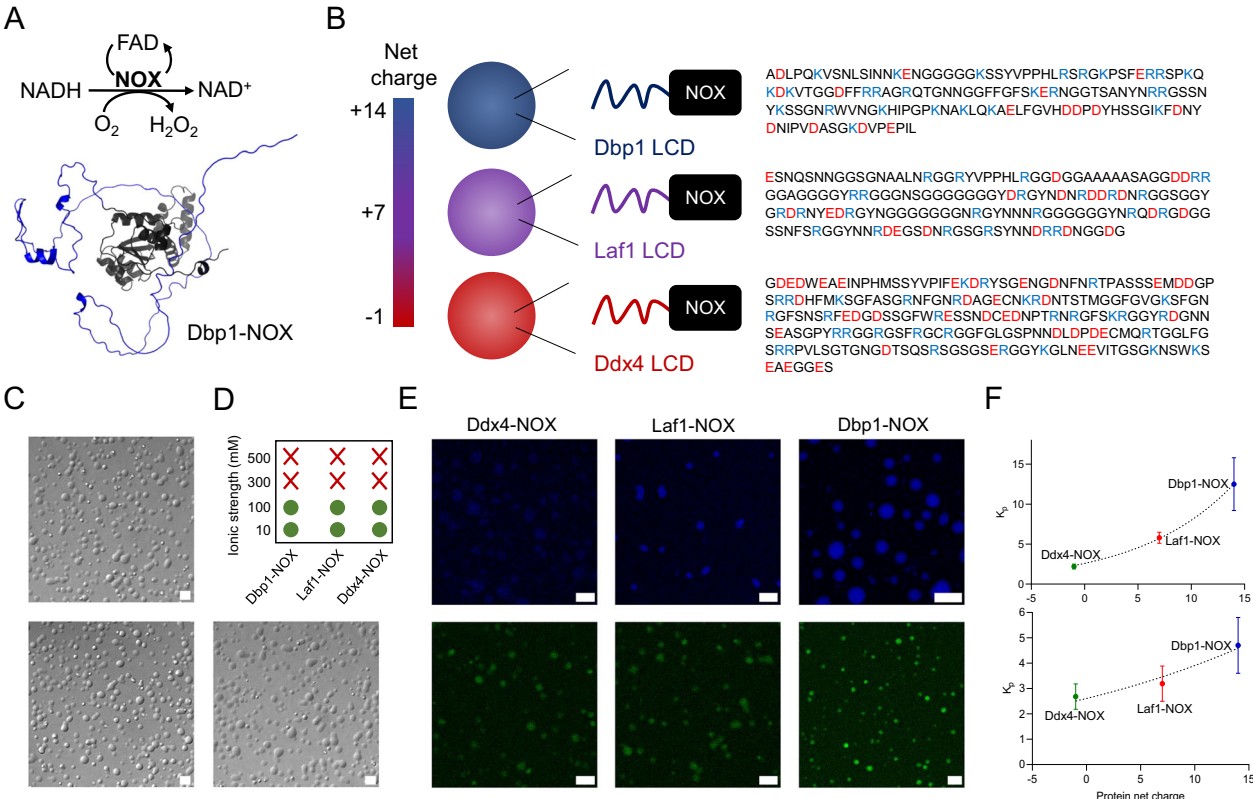

**Fig. 1 | Condensates based on different LCDs provide different microenvironments for NOX enzymatic activity. A** NOX catalyzes the oxidation of NADH to NAD$^+$ in the presence of oxygen, using FAD as a cofactor. AlphaFold[55] structure of Dbp1-NOX, showing the unstructured Dbp1 LCD (blue color) and the globular NOX enzyme (black color). **B** LCDs derived from the DEAD-box proteins Dbp1, Laf1, and Ddx4 are fused to the N-terminus of the NOX enzyme, creating chimeric proteins with different net charges, which are represented by the bar. The amino acid sequence of each LCD is shown on the right. Basic residues (Arg and Lys) are highlighted in blue, and acidic residues (Asp and Glu) are highlighted in red. **C** Representative phase-contrast microscopy images of solutions of 5 μM Ddx4-NOX (top), Dbp1-NOX (bottom, left), and Laf1-NOX (bottom, right) in 25 mM Tris, 20 mM NaCl, pH 7.5 showing the presence of micron-sized condensates when the enzyme is fused to LCDs. The scale bar represents 5 μm. The protein concentration of 5 μM was selected to facilitate visualization of the condensates under the confocal

microscope. Each experiment was repeated three times independently with similar results. **D** Phase separation is abolished at high salt concentration. Green circles and red cross indicate the presence and absence of phase separation, respectively. **E** Representative fluorescence confocal microscopy images showing the recruitment of NADH (top, blue fluorescence) and FAD (bottom, green fluorescence) in condensates of Ddx4-NOX (left), Laf1-NOX (middle) and Dbp1-NOX (right). NADH and FAD were added individually in two distinct experiments to avoid interference with the rapid reaction. The scale bar represents 5 μm. Each experiment was repeated three times independently with similar results. **F** Partitioning ($K_p$) of substrate (top) and cofactor (bottom) as a function of the net charge of the different fusion proteins. The dot lines represent exponential correlations and are a guide to the eyes only. $n = 3$ independent experiments. Data were presented as mean values ± SEM.

**Table 1 | Recruitment and concentration of NOX enzyme in the different condensates**

|  | Total volume fraction of the dense phase (%) | Amount of enzyme recruited in the dense phase (%) | NOX concentration in the dense phase (mM) | NOX concentration in the dilute phase (nM) |
|---|---|---|---|---|
| Dbp1-NOX | 0.015 ± 0.003 | 70 ± 5 | 4.6 | 300 ± 30 |
| Laf1-NOX | 0.016 ± 0.004 | 72 ± 7 | 4.5 | 280 ± 70 |
| Ddx4-NOX | 0.014 ± 0.006 | 71 ± 1 | 5.1 | 290 ± 10 |

We show that only heterogeneous systems with condensates formed with the most cationic LCD exhibit an increase in enzymatic rate. This finding suggests that the observed enhancement is due to a specific local environment generated by the condensates, which can modulate enzymatic reactions by acting as distinct effective solvents compared to the surrounding solution. Moreover, we demonstrate that the increase in enzymatic activity occurs across different length scales of the condensates, and is more pronounced in nanoclusters compared to the micron-sized condensates.

## Results

### Condensate microreactors with different local environments

We generated three different fusion proteins by conjugating the enzyme NOX with LCDs obtained from three different proteins belonging to the DEAD-box family of RNA-binding proteins (namely, Dbp1, Laf1, and Ddx4). The sequences of the chimeric proteins are shown in the Suppl. Fig. 1. The fraction of the intrinsically disordered regions in NOX chimeric proteins (30–50%) is comparable to biological RNA-binding proteins (e.g., hnRNPA1, FUS, and TDP-43). As the three LCDs have different amino acid sequences (Fig. 1B), they are expected to generate condensates with different physicochemical properties. This approach has the key advantage to enable the comparison of the same enzyme in different condensates characterized by different environments.

Specifically, the selected LCDs have drastically different net charges at physiological pH (−4 for Ddx4, +4 for Laf1, and +11 for Dbp1) (Fig. 1B). The Net Charge Per Residue (NCPR) are −0.017, +0.024, and +0.071 for Ddx4, Laf1, and Dbp1, respectively. The net charges of the LCD-NOX fusion proteins follow a similar trend, being −1, +7, and +14 for Ddx4-NOX, Laf1-NOX, and Dbp1-NOX, respectively.

We first confirmed the ability of the fusion proteins to undergo phase separation in the reference buffer conditions (25 mM Tris, 20 mM NaCl, pH 7.5). While the unconjugated NOX enzyme remained soluble under all investigated conditions, all fusion proteins formed condensates as revealed by confocal microscopy (Fig. 1C and Suppl. Fig. 2).

The phase separation of the selected LCDs is driven by a combination of intermolecular interactions, which include electrostatic interactions mediated by charge patterning as well as additional interactions induced by arginine and aromatic residues[9,39–41]. Our condensates could be dissolved by increasing the salt concentration above 300 mM (Fig. 1D), as previously reported for artificial condensates based on LCDs derived from DEAD-box proteins[29,30].

We next determined the recruitment of the different LCD-NOX fusion proteins into the different condensates by separating the dilute and dense phase by centrifugation and measuring the protein concentration in the dilute phase by size exclusion chromatography (SEC). We conducted experiments using a total enzyme concentration of 1 μM, chosen based on preliminary experiments to monitor the reaction within an optimal time scale of minutes while facilitating phase separation. Condensates obtained with different LCDs exhibited similar recruitment of the enzyme, being 70 ± 5%, 72 ± 7%, and 71 ± 1% for the Dbp1-NOX, Laf1-NOX, and Ddx4-NOX, respectively (Table 1).

The total volume fraction ($\Phi$) of the dense phase, measured by confocal microscopy, was also similar for all condensates (0.015%) (see

Table 1 and Methods). As a consequence, considering the simple mass balance equation $C_{tot} = C_{dense}\Phi + C_{dil}(1 − \Phi)$, where $C_{dense}$ and $C_{dil}$ are the protein concentration in the dense and dilute phase, respectively, $C_{tot}$ is the total protein concentration, and $\Phi$ indicates the volume fraction of the dense phase, the different condensates exhibited a similar concentration of enzyme in the dense phase of ~4.5–5 mM (Table 1).

Overall, the three different LCDs trigger the formation of protein condensates with comparable stimulus responsiveness, enzyme recruitment, size distribution, and total volume fraction. Yet, since both the substrate NADH and the cofactor FAD carry two negative charges at physiological pH, we expected that the recruitment of these small molecules would be largely different in the three condensates characterized by different net charges. To avoid any influence of the rapid reaction, we measured the partitioning of NADH and FAD independently by confocal microscopy. Specifically, we recorded their intrinsic fluorescence signal at 410–440 nm and 510–530 nm for NADH and FAD, respectively (Fig. 1E), and calculated the partition coefficient ($K_p$) as the ratio of fluorescence intensity inside and outside the condensates.

For the substrate, we measured a partition coefficient ($K_p$) of 12.5 ± 3.3, 5.8 ± 0.7, and 2.2 ± 0.3 for the Dbp1-NOX, Laf1-NOX, and Ddx4-NOX, respectively. These values increase with the net charge of the fusion proteins (Fig. 1F), confirming that electrostatic interactions are the main driving force for substrate partitioning. Moreover, the measured partition coefficients are in good agreement with previous results reported in the literature for the uptake of NADH into cationic condensates[42].

The cofactor partitioning coefficient ($K_p$) was lower than the substrate (Fig. 1F), despite following a similar trend ($K_p = 2.7 ± 0.5$, 3.2 ± 0.7, 4.7 ± 1.1 for Ddx4-NOX, Laf1-NOX, and Dbp1-NOX, respectively). The partitioning of molecules into the condensates was confirmed by the overlap between the fluorescence and bright-field images for Dbp1-NOX condensates (similar results were obtained for the other LCD-NOX constructs) (Suppl. Fig. 3). These findings confirm the increased uptake of the negatively-charged substrate and cofactor by the most cationic condensates.

### Condensates with suitable local environment modulates NOX enzymatic activity

We next analyzed the activity of the NOX enzyme within the different condensates, and compared the results with unconjugated NOX. To this end, we measured the substrate consumption over time by recording its characteristic absorbance signal at 340 nm, and evaluated the initial reaction rate. Representative profiles are reported in Fig. 2A. In the presence of condensates, this rate represents the average contribution from the dilute and the dense phase. To confirm the absence of any potential effect of the condensates on the absorbance signal at 340 nm, we monitored the absorbance of NADH in the presence of Dbp1-NOX condensates in the absence of reaction (i.e., in the absence of FAD cofactor), observing a negligible change of the signal over time (Suppl. Fig. 4).

While the initial rate measured with condensates composed of Laf1-NOX and Ddx4-NOX was similar to the homogeneous NOX solution, we observed a faster rate in the Dbp1-NOX heterogeneous system (Fig. 2A, B).

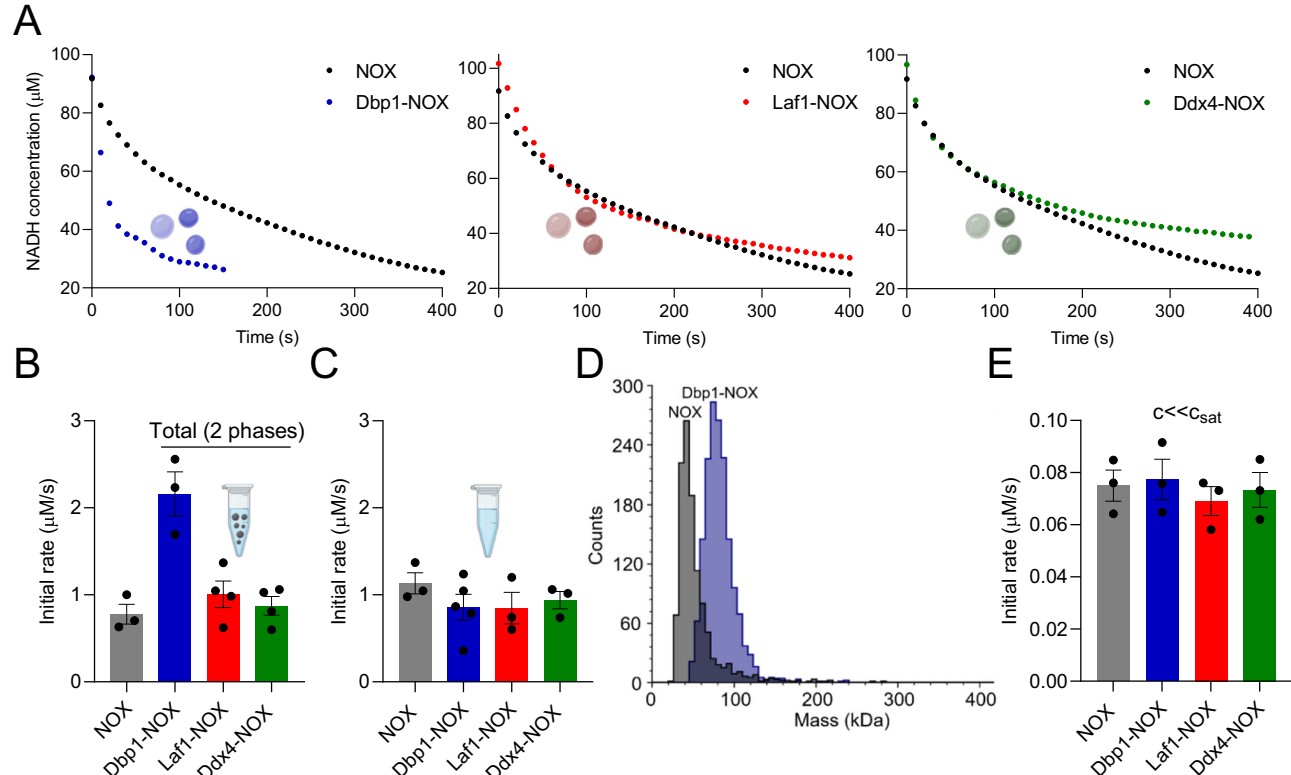

**Fig. 2 | Condensates formed by different chimeric proteins modulate NOX enzymatic activity. A** Representative profile of the reaction progress characterized by a decrease in the NADH absorbance at 340 nm for the homogeneous solution (black symbols, NOX) and the heterogeneous system composed of condensates and the dilute phase (blue symbols, Dbp1-NOX, red symbols, Laf1-NOX and green symbols, Ddx4-NOX). Proteins were diluted to 1 μM in 25 mM Tris, 20 mM NaCl pH 7.5. **B** Initial rates measured for the NOX homogeneous system (gray bar) and for the heterogeneous systems ("Total (2 phases)") at 1 μM in 25 mM Tris, 20 mM NaCl pH 7.5. $n \geq 3$ independent experiments. Data were presented as mean values ± SEM. **C** Initial rates of 1 μM protein solutions at high ionic strength (500 mM NaCl), where a single homogeneous phase was observed for all proteins. $n \geq 3$ independent experiments. Data were presented as mean values ± SEM. **D** Mass distribution of 20 nM NOX and Dbp1-NOX solutions at low ionic strength (20 mM NaCl) analyzed by single-molecule mass photometry. Both peaks correspond to the dimeric forms of the protein: 45 kDa for NOX and 79 kDa for Dbp1-NOX. **E** Initial rates of the homogeneous systems at 20 nM enzyme ($c \ll c_{sat}$) and low ionic strength (20 mM NaCl). $n = 3$ independent experiments. Data were presented as mean values ± SEM. Created with BioRender.com.

To rule out an effect of the conjugation of the Dbp1 LCD on the enzymatic activity, we measured the reaction rate in homogeneous solutions of conjugated and unconjugated enzyme at 500 mM salt concentration, where no condensates could be detected (Suppl. Fig. 5). As shown in Fig. 2C, the initial rate was comparable for all constructs.

Moreover, we measured the enzymatic rate at very low enzyme concentration (20 nM) at low salt concentration (20 mM NaCl). Single-molecule mass photometry analysis indicated that under these conditions, Dbp1-NOX is mainly present as a dimer (Fig. 2D). Also, in this case, all constructs exhibited the same initial rate (Fig. 2E). Overall, these results show that conjugation of NOX with the LCDs does not significantly modify the enzymatic activity, in agreement with previous findings with a kinase[29,30].

The increased rate observed with the Dbp1-NOX system stems, therefore, from the enhancement of the enzymatic activity triggered by condensation. Since the different condensates have similar volume fraction, as well as amount and concentration of enzyme, this enhancement is due to the local microenvironment generated by the positively charged Dbp1 LCD.

**Protein nanoclusters in the dilute phase increase the enzymatic activity of NOX**

To further confirm the enhancement of enzymatic activity due to condensation, we removed the condensates by centrifugation and quantified the reaction rate in the dilute phase. The initial rate measured in the dilute phase of Laf1-NOX and Ddx4-NOX was lower than the control homogeneous solution of 1 μM NOX (Fig. 3A). The reduction of 44 and 40% for Laf1-NOX and Ddx4-NOX, respectively, was consistent with the ~30% decrease of enzyme concentration in the dilute phase due to recruitment in the condensates. Moreover, these initial rates were comparable to the value measured in a control NOX homogeneous solution at 280 nM, which corresponds to the enzyme concentration in the dilute phase of the heterogeneous system (Fig. 3A). These results further confirm that conjugation of Laf1 or Ddx4 LCD with NOX does not alter enzymatic activity in either the dilute or the dense phase.

The enzymatic rate in the dense phase ($r^{II}$) is still locally much higher than the dilute phase ($r^{I}$) due to the increase in enzyme concentration, but this effect is compensated by the low volume fraction of the dense phase ($\Phi$). As a result, the overall reaction rate of the two-phase system ($r = r^{I}(1 - \Phi) + r^{II}\Phi$) is the same of the control homogeneous solution.

In analogy with Laf1- and Ddx4-NOX, also for Dbp1-NOX, the reaction rate in the dilute phase was lower than the total reaction rate of the two-phase system due to the recruitment of the enzyme in the dense phase (Fig. 3A). However, in contrast to Laf1- and Ddx4-NOX, for Dbp1-NOX the rate in the dilute phase was comparable to the rate in the control homogeneous solution of 1 μM NOX, despite the concentration of the enzyme in the dilute phase being ~30% of the value in the homogenous solution. These results indicate an increase in the activity of the Dbp1-NOX in the dilute phase. The experiments

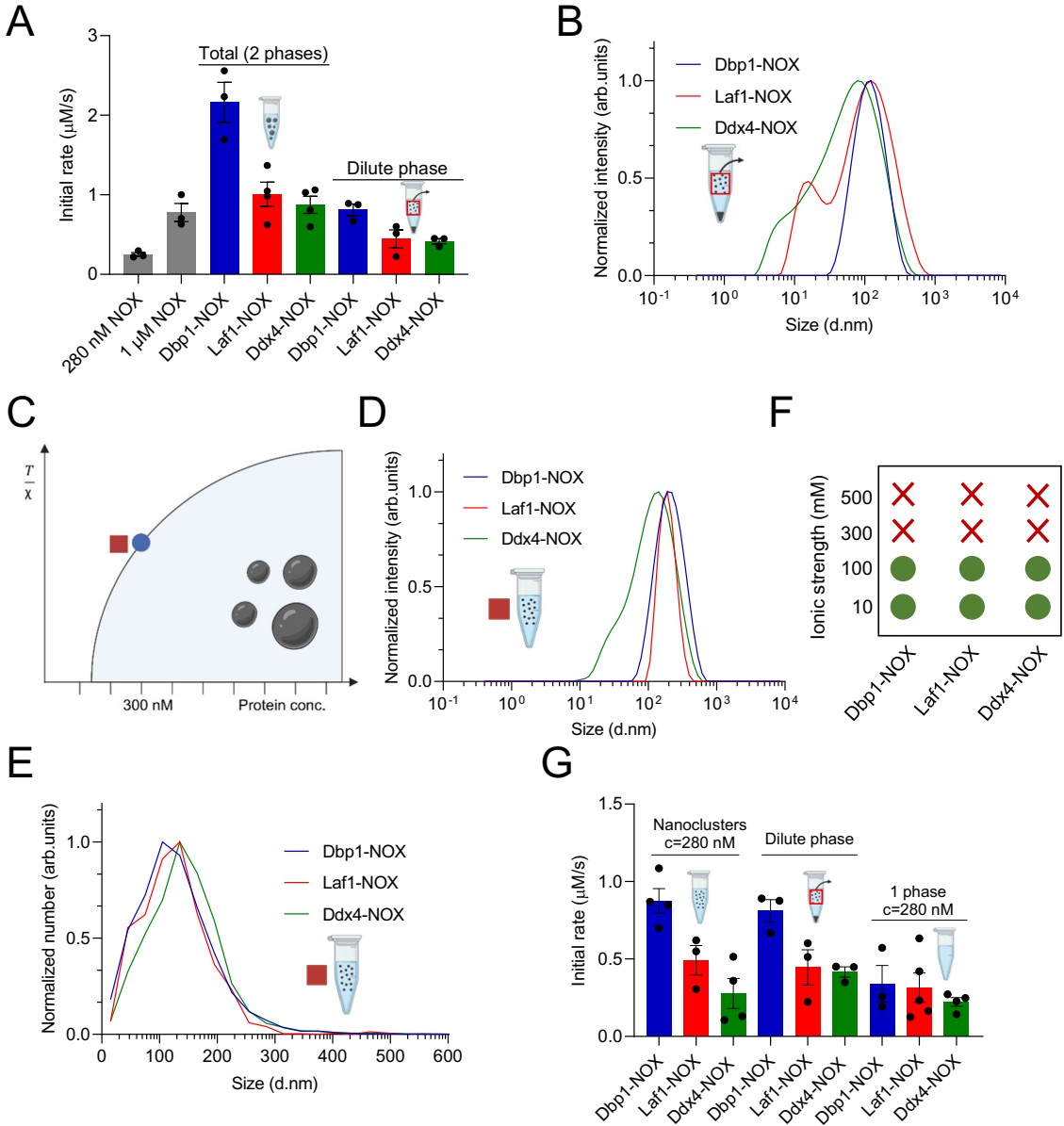

**Fig. 3 | Modulation of enzymatic activity in nanoclusters. A** Initial rates of homogeneous NOX solutions at 280 nM and 1 μM (gray), of heterogeneous systems formed by the different fusion proteins at 1 μM protein concentration ("Total (2 phases)"), and of the dilute phase in equilibrium with the dense phase in the heterogeneous systems ("Dilute phase"). $n \geq 3$ independent experiments. Data were presented as mean values ± SEM. **B** Size distribution of the dilute phase after removing micron-sized condensates by centrifugation measured by dynamic light scattering, showing the presence of nanoclusters. **C** Schematic phase diagram showing the protein concentration at which nanoclusters were formed (280 nM, red square), and the $c_{sat}$ (300 nM, blue circle). The Y-axis represents temperature normalized by the interaction coefficient ($\chi$). **D, E** Size distribution of 280 nM protein solutions in 25 mM Tris, 20 mM NaCl, pH 7.5 measured by dynamic light scattering (**D**) and nanoparticle tracking analysis (**E**), showing the presence of nanoclusters. **F** Presence and absence of nanoclusters as detected by dynamic light scattering, indicated by green circles and red cross, respectively. The formation of the clusters is abolished at salt concentrations larger than 300 mM. **G** Initial rates of the solutions at 280 nM in 25 mM Tris at low salt (20 mM NaCl, where nanoclusters are observed, "Nanoclusters c = 280 nM") and high salt (500 mM NaCl, where the solution is homogeneous, "1 phase c = 280 nM"). The initial rates of the dilute phase in equilibrium with the dense phase in the heterogeneous system reported in panel (**A**) ("Dilute phase") are also shown in this panel (**G**) for comparison. $n \geq 3$ independent experiments. Data were presented as mean values ± SEM. Created with BioRender.com.

discussed in the previous paragraph (Fig. 2C, E) demonstrated that the conjugation per se does not affect enzymatic activity.

We speculated that the increase in activity in the dilute phase could be due to the presence of clusters in the nanometer range. To test this hypothesis, we performed dynamic light scattering (DLS) and nanoparticle tracking analysis (NTA) of the dilute phase, which revealed the presence of clusters with an average diameter of ~100 nm for all chimeric constructs (Fig. 3B and Suppl. Fig. 6). These observations are compatible with the presence of pre-percolation clusters

associated with phase separation coupled to percolation (PSCP) process or with nanocondensates resisting coalescence[43–45].

Based on these observations, we next aimed at determining the effect of the size of the condensate on the enzymatic activity. We prepared samples at protein concentrations close to the saturation concentration required to form micron-sized condensates ($c_{sat}$), where smaller condensates are expected to form (Fig. 3C and Suppl. Figs. 7, 8). Dynamic light scattering analysis revealed the presence of clusters with a diameter distribution in the range of 100–200 nm (Fig. 3D),

which is similar to the size distribution of the clusters in the dilute phase of the heterogeneous system (Fig. 3B). Similar size distributions were measured by NTA (Fig. 3E). Furthermore, nanoclusters at a concentration of 280 nM remained colloidally stable and did not grow into micron-scale condensates during the time scale of the reaction, as monitored by dynamic light scattering (Suppl. Fig. 9). To estimate the amount of protein recruited into the nanoclusters, solutions of 280 nM Dbp1-NOX were subjected to ultracentrifugation and the dilute phase was subsequently analyzed by SEC. $40 \pm 7\%$ of the Dbp1-NOX protein was recruited into the nanoclusters. As for the assemblies observed in the dilute phase, also these nanoclusters could represent pre-percolation clusters associated with phase separation coupled to the percolation (PSCP) process or nanocondensates resisting coalescence[43–45].

We analyzed the interactions driving the formation of these nanoclusters by measuring their formation at different salt concentrations by dynamic light scattering (Fig. 3F). In analogy to the micron-sized condensates (Fig. 1D), the nanoclusters dissolved at salt concentrations higher than 300 mM, indicating that their formation is mediated by intermolecular interactions that are similar to the micron-sized condensates.

At the salt concentration of 500 mM, the enzymatic activity was similar for all constructs and comparable with the NOX homogeneous system (Fig. 3G), further confirming the absence of nanoclusters at this high salt concentration and the lack of effect of the conjugation of the LCD on enzymatic activity.

In contrast, the initial rates measured in the solutions at 280 nM protein concentration at low salt concentrations, where nanoclusters are present, show an increase of the enzymatic activity for Dbp1-NOX in comparison with the other LCDs. Moreover, the observed values are consistent with the activity of the dilute phase in equilibrium with the dense phase in the heterogeneous system, where a comparable amount of nanoclusters is present (Fig. 3G). These results further confirm that assemblies of Dbp1-NOX exhibiting a local increase of substrate and cofactor, either in the form of nanoclusters or micron-sized condensates, are responsible for the observed increased enzymatic activity.

## Enhancement of enzymatic activity by local environment occurs across different length scales and is larger in nanoclusters

To analyze more quantitatively the effect of condensation on Dbp1-NOX activity, we measured the enzymatic rate at different initial substrate concentrations for unconjugated NOX and for Dbp1-NOX nanoclusters and micron-sized condensates (formed at 280 nM and 1 μM protein concentration, respectively) (Fig. 4).

In the range of substrate concentrations from 50 to 200 μM, the data could be described by a simple Michaelis–Menten kinetics (Fig. 4). An inhibitory effect of the substrate and a more complex behavior was observed at higher NADH concentrations (Suppl. Fig. 10), which were therefore not considered for the analysis.

In the presence of Dbp1-NOX micron-sized condensates we observed a threefold increment of the apparent $k_{cat}$ ($3.87 \pm 0.60$ s$^{-1}$) and a similar apparent $K_M$ ($89 \pm 34$ μM) in comparison with the homogeneous NOX solution ($k_{cat} = 1.12 \pm 0.21$ s$^{-1}$, $K_M = 81 \pm 36$ μM) at 1 μM enzyme (Fig. 4A). We note that, for the two-phase system, the $k_{cat}$ and $K_M$ parameters of the fitted Michaelis–Menten equation represent only apparent values that report on the average contribution of the dilute and dense phase.

Kinetic parameters measured in homogeneous NOX solutions at 280 nM were consistent with values measured at 1 μM enzyme ($k_{cat}$ of $1.66 \pm 0.17$ s$^{-1}$ and $K_M$ of $96 \pm 20$ μM). In analogy with the micron-sized condensates, also in the presence of Dbp1-NOX nanoclusters, we observed an increment in the apparent $k_{cat}$ of approximately threefold ($4.96 \pm 0.91$ s$^{-1}$) and negligible changes in the apparent $K_M$ ($103 \pm 40$ μM) (Fig. 4B).

The results show that the change in the overall enzymatic activity due to the local microenvironment of the Dbp1-NOX dense phase occurs across different length scales.

We further analyzed the enhancement of the reaction rate in the dense phase by deconvoluting the contribution of the individual phases to the average reaction rate. We started from the 280 nM solution, where nanoclusters are present. The average rate can be expressed as $r = r_1 \Phi_1 + r_2 \Phi_2$, where the indexes 1, and 2 indicate, respectively, the dimers in the dilute phase and the nanoclusters. $\Phi$ represents the volume fractions, which have been previously estimated experimentally. Assuming the same rate in the dilute and homogenous phase, and normalizing for the enzyme concentration in the two phases (see details in Methods), we estimated $r_{2,norm} = 6.1 \cdot r_{1,norm}$. Such increment could be due to either a mass action effect related to the partitioning of substrate and cofactors, or to a change in the kinetic parameters. Recent theoretical work highlights that the activity coefficient of client molecules in the dense phase can be significantly reduced compared to the dilute phase and can compensate for the local increase in concentration[46]. Changes in the total reaction rate (of both dilute and dense phase) compared to the homogeneous solution should therefore be attributed to changes in the kinetic parameters of the enzyme rather than mass action laws[46]. The absence of a mass action effect in our systems is further supported by the lack of enhancement observed in two out of three systems (Laf1-NOX and Ddx4-NOX), despite all systems locally concentrating substrates with different partitioning coefficients (Fig. 1F). In other words, the changes in the reaction rate are not proportional to the partitioning of substrate and cofactor. Moreover, measurements of the initial rates for the Dbp1-NOX heterogeneous system at different cofactor concentrations showed a nonlinear increase of activity with cofactor concentration, reaching a plateau at 200 μM (Suppl. Fig. 11).

Altogether, these observations point to an increase in the $k_{cat}$ in the dense phase as the reason behind the rate enhancement in the Dbp1-NOX system.

We note that in our system the enzyme is incorporated in the scaffold protein of the condensates and does not participate in the interactions driving phase separation, which are largely mediated by the LCDs. A key feature of our strategy is that the change in the activity coefficient of the enzyme in the dense phase is, therefore, negligible, and high enzymatic rates can be achieved inside the dense phase. The enzymatic reaction is always locally accelerated in the condensates compared to the surrounding dilute phase due to the larger concentration of the enzyme in the dense phase, even for the constructs that do not show a rate enhancement (Fig. 4C).

Finally, we evaluated the effect of the size of the condensates on the increase in enzymatic rate by estimating the contribution of the micron-sized condensates and the nanoclusters for the 1 μM Dbp1-NOX sample. The average reaction rate can be expressed as $r = r_1 \Phi_1 + r_2 \Phi_2 + r_3 \Phi_3$, where the indexes 1, 2, and 3 indicate, respectively, the dimers in the dilute phase, the nanoclusters, and the micron-sized condensates. Assuming the same rate in the nanoclusters in the solutions at 280 nM and 1 μM protein, and normalizing again for the enzyme concentration (see Methods), we obtained $r_{3,norm} = 2.1 \cdot r_{1,norm}$ which indicates a larger rate for the nanoclusters ($r_2 = 2.9 \cdot r_3$) compared to the micron-sized condensates. This result can be due to the presence of mass transfer limitations. To test this hypothesis, we estimated the characteristic diffusion time of a small molecule in the condensates by fluorescence correlation spectroscopy (FCS) (Suppl. Fig. 12). Considering a characteristic radius $L$ of 1 μm for the condensates, based on simple scaling analysis the measured diffusion coefficient $D$ of 3.8 μm$^2$/s corresponds to a characteristic diffusion time of $\tau_D = L^2/D$ of 263 ms. This time is comparable to the estimated characteristic time of the reaction $\tau_R = 1/k_{cat}$ of 258 ms (the Damköhler number, $Da = \tau_D/\tau_R$, is ~1), indicating the possible presence of mass transfer limitations. However, we cannot exclude that the larger rate

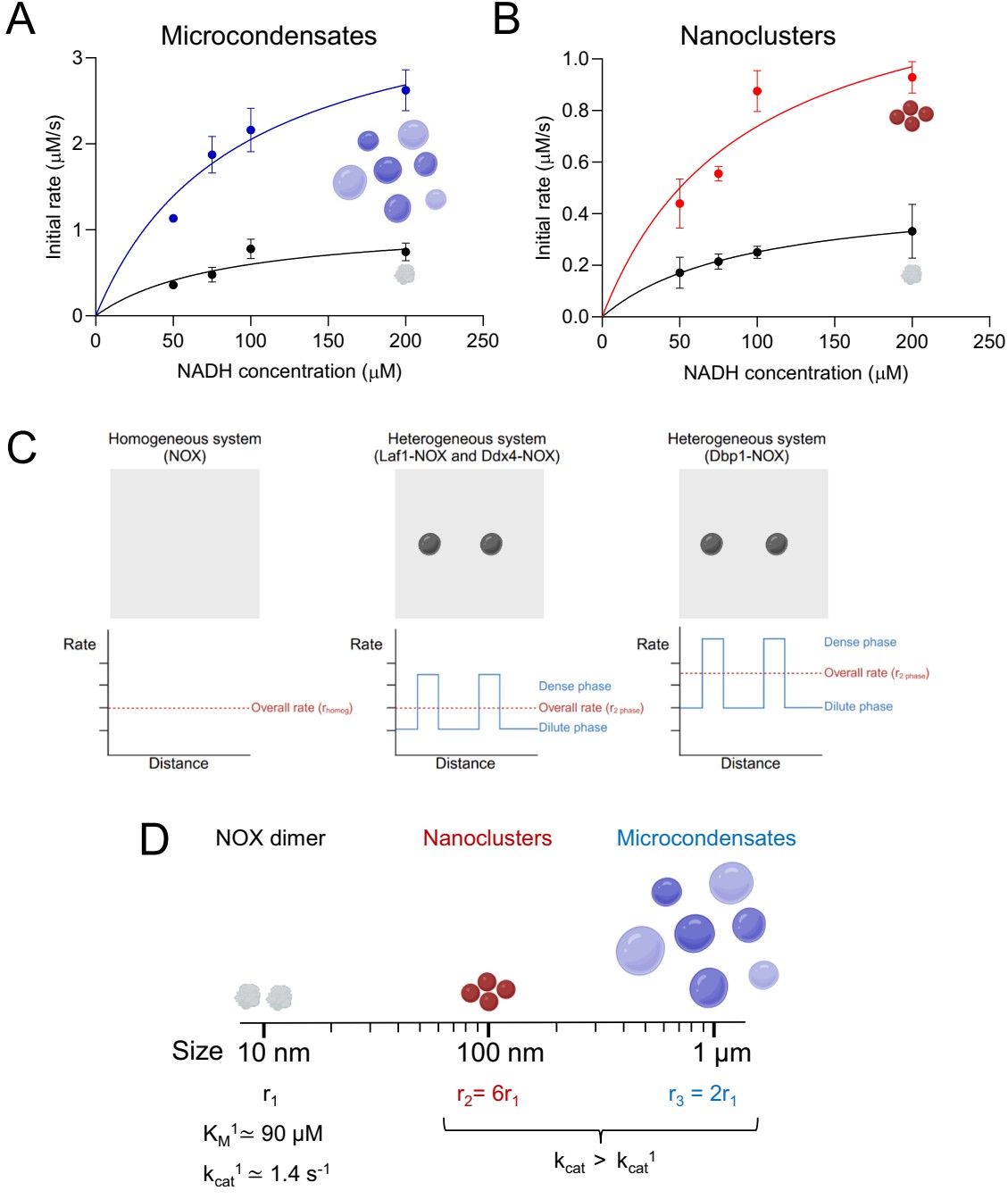

**Fig. 4 | Increase of NOX enzymatic activity in Dbp1-NOX microcondensates and nanoclusters. A**, **B** Michaelis–Menten plot of 1 μM (**A**) and 280 nM (**B**) NOX (black) and Dbp1-NOX (blue and red) in 25 mM Tris, 20 mM NaCl, pH 7.5. Symbols indicate experimental data and continuous lines in the model simulations. $n \geq 3$ independent experiments, except for one data point, which is derived from $n = 2$, and no error bars are derived. Data were presented as mean values ± SEM. **C** Schematic illustration of the initial rates in the dilute and dense phase (blue color) and of the overall initial rate of the entire system (red color) for the homogeneous NOX solution and LCD-NOX heterogeneous systems. **D** Schematic illustration showing the effect of protein condensation on the normalized rates across length scales. The sizes of 10 nm, 100 nm, and 1 μm indicate, respectively, the protein dimer, the nanoclusters, and the micron-sized condensates. Created with BioRender.com.

for smaller condensates is potentially due to also a higher reactivity of the enzyme at the interface of the condensates. Indeed, interfaces can affect protein conformations[47] and have been recently shown to modulate protein aggregation events[48–50].

## Discussion

We have incorporated the same NOX enzyme into condensates formed with different LCDs, each characterized by a distinct net charge. Through this approach, we generated condensates with different local microenvironments and investigated their impact on the enzymatic

activity. The conjugation of LCDs to NOX did not alter the activity of the enzyme in the absence of condensation.

We observed an increase in the enzymatic rate in condensates formed with the most cationic Dbp1 LCD, while condensation based on the other LCDs showed no effect. Since the enzyme concentration is similar in all condensates, these findings indicate that the increase in enzymatic rate is due to the distinct local microenvironment. Moreover, the variations in enzymatic rate among the different condensates do not correlate with the partitioning of substrate and cofactor. This observation, together with the arguments discussed above, suggests

that the enhancement of enzymatic rate results from a change in enzymatic activity within the dense phase rather than a mass action effect.

One possible explanation for this alteration could be the existence of specific interactions between the Dbp1 LCD and the NOX enzyme, which occur exclusively within the dense phase due to conformational changes in the condensates with respect to the dilute solution. Indeed, dimeric NOX features two anionic patches on its surface that could potentially engage in electrostatic interactions with the cationic Dbp1 LCD (Suppl. Fig. 13).

Oxygen ($O_2$) plays a crucial role in the enzymatic reaction catalyzed by NOX and could potentially also contribute to the rate enhancement observed for Dbp1-NOX. Different microenvironments within the condensates may also affect $O_2$ saturation levels. However, given the low water solubility of $O_2$, it is unlikely that condensates are saturated with $O_2$ under the tested quiescent conditions.

In any case, the observed enhancement of enzymatic rate is attributed to the specific local microenvironment of the condensates formed with one single LCD. Biomolecular condensates based on the disordered domain of the helicase Ddx4 have previously been shown to facilitate the melting of DNA double helices by acting as an organic-like solvent[51]. Our results suggest that this solvent-like effect can underlie the acceleration of other reactions mediated by condensates, which effectively function as distinct solvents compared to the surrounding solution.

Importantly, we observed the influence of the local environment of the condensate on the enzymatic rate across a wide size range, from nanometers to microns. This result indicates that emergent properties of condensates (in this case, the enhancement of enzymatic rate) can manifest within assemblies as small as nanometers, a scale which is relevant for many cellular organelles.

Moreover, our analysis revealed that the rate enhancement is larger in smaller condensates, suggesting that the increase in enzymatic activity is not only relevant but may potentially be even more pronounced in nanometer-sized cellular condensates.

In addition to implications for biology, our findings are relevant for the design of synthetic enzymatic condensates. Here we demonstrated the potential of the approach with NOX enzymes, which are widely employed as cofactor regeneration systems in various relevant industrial processes such as pharmaceutical synthesis[52,53]. These condensate protein materials can represent an alternative to traditional protein immobilization methods, as they bypass conjugation steps and maintain the enzyme in a biological-like environment.

## Methods

### Protein expression and purification
Genes encoding the different fusion proteins were codon optimized for expression in *Escherichia coli* (*E. coli*), synthetized, and cloned into the pET-15b vector by Genewiz (NJ, US). We fused the NADH oxidase from *Thermus thermophilus* (NOX) with the LCD of Dbp1 from *Saccharomyces cerevisiae* (from residue 1 to 155), Laf1 from *Caenorhabditis elegans* (from residue 1 to 168), and Ddx4 from human (from residue 1 to 236). *E. coli* BL21-GOLD (DE3) competent cells were transformed with the corresponding plasmids encoding NOX, Ddx4-NOX, Dbp1-NOX, and Laf1-NOX, and grown aerobically in Luria-Bertani broth (LB) medium supplemented with 100 μg/mL ampicillin. Protein expression was induced at $OD_{600}$ of 0.6 with 0.5 mM isopropyl ᴅ-thiogalactopyranoside (99%, PanReac AppliChem) for 16 h at 20 °C.

NOX enzyme was purified using a heat shock method[37]. Dbp1-NOX, Laf1-NOX, and Ddx4-NOX proteins were purified by a combination of heat shock and immobilized metal ion affinity chromatography (Chelating Sepharose, GE Healthcare) according to a standard protocol[29,30]. The final purity was confirmed by SDS-PAGE, and protein concentration was determined by measuring absorbance at 280 nm.

### Analysis of phase separation
The phase diagrams of the different fusion proteins were obtained by analyzing 1 μM protein solutions by bright-field microscopy (Eclipse Ti-E, Nikon) using a 60x oil objective (FI Plan Apo Lambda NA 1.4, Nikon). To this aim, the fusion proteins were dissolved from stock solutions into 25 mM Tris buffer, 10 mM NaCl, pH 7.5, and allowed to phase separate for 10 min. After analyzing the presence of condensates, the salt concentration was progressively increased to 100 mM, 300 mM, and 500 mM, and images were acquired after 10 min equilibration after each change of salt concentration.

To estimate the $c_{sat}$ of the different fusion proteins, samples were prepared at different protein concentrations (from 100 nM to 1 μM) at 25 mM Tris, 20 mM NaCl, pH 7.5, and allowed to equilibrate for 10 min. The presence of condensates was analyzed by bright-field microscopy (Eclipse Ti-E, Nikon) using a 60x oil objective (FI Plan Apo Lambda 60× Oil, Nikon).

All the measurements were performed in a 384-well plate (Matri-Plate 384-Well Plate, Glass Bottom, Brooks).

### Size exclusion chromatography (SEC)
Size exclusion chromatography (SEC) was applied to determine the enzyme concentration in the dilute phase. A solution containing 1 μM fusion protein in 25 mM Tris, 20 mM NaCl pH 7.5 was incubated for 45 min at room temperature. The dilute and dense phase were separated by centrifugation at $8000 \times g$ for 30 min. The supernatant was carefully removed and loaded on a Superdex 200 Increase 5/150 GL (Cytiva) assembled on an Agilent 1200 Series HPLC coupled to a fluorescence detector (Agilent). The elution profiles were monitored using the intrinsic tryptophan fluorescence of the fusion proteins. The amount of protein in the supernatant was compared to a reference 1 μM protein solution at high salt concentration (25 mM Tris, 500 mM NaCl, pH 7.5) in order to estimate the percentage of enzyme recruited in the dense phase.

The percentage of recruited protein into the nanoclusters of Dbp1-NOX was determined by following the same protocol, but the dense phase was removed by ultracentrifugation at $180,000 \times g$ for 120 min at room temperature.

### NOX activity assay
The enzymatic activity of NOX and the different fusion proteins was determined by monitoring the decrease in the absorbance of the substrate NADH at 340 nm in a CLARIOstar plus plate reader (BMG Labtech). Samples contained 1 μM enzyme, NADH in the concentration range from 50 to 500 μM, and 100 μM FAD in 25 mM Tris, 20 mM NaCl, pH 7.5. For the analysis of homogeneous solutions at high salt, proteins were dissolved at 1 μM in the presence of 100 μM NADH and 100 μM FAD in 25 mM Tris, pH 7.5, and 500 mM NaCl.

To quantify the activity of the dilute phase, the dense phase was removed by centrifugation at $8000 \times g$ for 30 min. About 100 μM NADH and 100 μM FAD were rapidly added to the dilute phase, and the activity was measured with the same procedure described above.

The activity of protein solutions at 280 nM at low and high ionic strength, and 20 nM at low ionic strength was measured in the presence of 100 μM NADH and 100 μM FAD.

The influence of condensate light scattering in the NADH absorbance was assessed by recording the absorbance at 340 nm of a solution containing 1 μM Dbp1-NOX and 100 μM NADH in 25 mM Tris, 20 mM NaCl, pH 7.5 during 30 min.

To quantify the activity of Dbp1-NOX at different cofactor concentrations, samples containing Dbp1-NOX were incubated in the presence of 100 μM NADH and FAD in the concentration range from 25 μM to 500 μM in 25 mM Tris, 20 mM NaCl, pH 7.5.

Michaelis−Menten kinetics were evaluated at different substrate concentrations and 100 μM FAD. The kinetic parameters ($K_M$ and $k_{cat}$)

were calculated by fitting the curve to the Michaelis–Menten equation in GraphPad Prism 8.

## Confocal microscopy

Phase separation and recruitment of client molecules were analyzed using a confocal microscope (Leica TCS SP8; Leica Application Suite X (LAS X) software, version 1.0) equipped with a 63× NA 1.4 oil objective (Leica). The NADH intrinsic fluorescence signal was measured using an excitation laser at 405 nm and recording the fluorescent signal at 410–440 nm. The FAD intrinsic fluorescence intensity was measured using an excitation laser at 458 nm and recording the emission signal at 510–530 nm. Samples were analyzed in 384-well plates (MatriPlate 384-Well Plate, Glass Bottom, Brooks).

The partitioning of both molecules into the condensates was calculated by the ratio of the fluorescence intensity inside and outside the condensates.

For the calculation of the volume fraction (Φ), condensates composed of the different fusion proteins were deposited on the bottom of the well, and a Z-stacking acquisition was performed each 0.2 μm. The different stacks were analyzed using Fiji and an in-house script able to calculate the Φ considering different parameters. Measurements were performed in triplicate and in different zoom factors.

## Dynamic light scattering (DLS)

Dynamic light scattering (DLS) analysis was performed on a Zetasizer Nano-ZS (Malvern) at 25 °C. For the analysis of the dilute phase in equilibrium with the condensates, heterogeneous solutions at 1 μM enzyme concentration were centrifuged at 8000 × g for 30 min to separate the dilute and the dense phase.

For the analysis of cluster formation below the $c_{sat}$, the different fusion proteins were prepared at 280 nM protein concentration and low ionic strength (20 mM NaCl). To follow the growth of clusters during time, the intensity autocorrelation function was recorded at different time points at 25 °C.

To analyze the salt dependence of cluster formation, 280 nM protein solutions were prepared at low ionic strength and incubated for 5 min. Subsequently, the salt concentration was progressively increased to 100, 300, and 500 mM, and the presence of clusters was analyzed using DLS.

## Nanoparticle tracking analysis (NTA)

NTA measurements of protein clusters in the dilute phase in equilibrium with condensates, and at a protein concentration below $c_{sat}$, were performed using a ZetaView instrument equipped with a CMOS camera and a 405 nm laser (Particle Metrix). The instrument was calibrated according to the manufacturer's protocol using polystyrene nanoparticle standards. For the analysis of cluster formation below $c_{sat}$, samples were prepared at 280 nM protein concentration and low ionic strength (20 mM NaCl), and injected into the sample chamber using a 1 mL syringe until the chamber was completely filled. For the analysis of the dilute phase in equilibrium with the condensates, heterogeneous solutions at 1 μM enzyme concentration were centrifuged at 8000 × g for 30 min to separate the dilute and the dense phase. Subsequently, samples were injected into the chamber using a 1 mL syringe. The chamber was washed with a buffer between two sample measurements until no particles were visible. Video acquisition was performed at 11 different positions with the shutter set to 100 and a high frame rate. The sensitivity was adjusted for each sample to a scattering intensity below 20. More than 1000 traces were analyzed per sample. Data analysis was performed using the ZetaView analysis software (ZetaView 8.04.02 SP2).

## Single-molecule mass photometry (SMMP)

SMMP measurements were performed on a Refeyn One$^{MP}$ mass photometer (Refeyn), calibrated according to the manufacturer's protocol.

Before sample measurements, the buffer was freshly prepared and filtered. Buffer cleanliness was checked before each measurement to ensure the lack of any particle generating optical contrast. Solutions of 20 nM NOX and Dbp1-NOX in low ionic strength buffer (20 mM NaCl) were measured over a time frame of 1 min at 25 °C. SMMP data analysis was performed using DiscoverMP software (Refeyn).

## Kinetic modeling

The overall reaction rate in a three-phase system can be expressed as $r = r_1\Phi_1 + r_2\Phi_2 + r_3\Phi_3$, where the indexes 1, 2, and 3 indicate, respectively, the dimers in the dilute phase, the nanoclusters, and the micron-sized condensates. $\Phi$ represents the volume fractions, which have been estimated experimentally. The volume fraction of the nanoclusters was estimated assuming the same enzyme concentration in the nanoclusters and in the micron-sized condensates. We further assumed that the rate in the nanoclusters was the same in the solutions at 280 nM and 1 μM protein concentration. Since the enzyme does not participate in the interactions driving phase separation, we did not consider any change in its activity coefficient. Therefore, assuming a simple Michaelis–Menten kinetics in all phases, the reaction rate is proportional to the enzyme concentration $E$: $r = k_{cat} \cdot E \cdot S/(K_M + S)$, while for the substrate, the activity $S$ should be considered. By dividing the reaction rate $r$ by the enzyme concentration $E$ ($E_{dense\ phase} = 2.6 \times 10^4 \cdot E_{dilute\ phase}$), we can compare the rates in the nanoclusters and condensates with the rate of the dimeric enzyme at the same enzyme concentration. Changes in the reaction rate can occur due to either changes in $k_{cat}$ or combined changes in $K_M$ and the activity of the substrate $S$. The arguments discussed in the text point to a change in $k_{cat}$ rather than a mass action effect. The values used for the kinetic modeling are shown in Suppl. Table 1.

## Fluorescence correlation spectroscopy (FCS)

FCS experiments were performed on an inverted confocal fluorescence microscope (Leica SP8 STED, Leica Application Suite X software, version 1.0) equipped with an HC PL APO CS2 63×1.2 NA water immersion objective with software-controlled correction collar (Leica) and a hybrid detector for single molecule detection (HyD SMD). The confocal volume was calibrated using Atto-565 NHS Ester (Diffusion coefficient = 400 μm²/s), which yielded an effective volume ($V_{eff}$) of 0.3 ± 0.01 fl ($n = 3$ measurements) and a focal volume height-width ratio, K = 5.1 ± 0.2 ($n = 3$ measurements). The samples were excited with a 555 nm laser (from a White Light Laser at 80 MHz repetition frequency), and the fluorescence emission was collected at a wavelength range of 570–610 nm. The pinhole size was 100 μm. The diffusivity of the Atto-565 inside the Dbp1-NOX condensates was extracted from the autocorrelation curve assuming no distortion of the confocal volume inside the condensates, as previously done[54]. The autocorrelation curves were fitted to a model assuming a single diffusing component, and the extracted diffusion coefficient is 3.8 ± 0.5 μm²/s ($n = 4$ measurements).

## Reporting summary

Further information on research design is available in the Nature Portfolio Reporting Summary linked to this article.

## Data availability

The data that support the findings of this study are available from the corresponding author upon request. Source data are provided as a Source Data file. The PDB code used in this study is 1NOX. Source data are provided with this paper.

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

## Acknowledgements

The authors thank the Riek group (ETH Zurich) for assistance with the mass photometry analysis, Dr. Umberto Capasso Palmiero for scientific discussions, and the Bringing Materials to Life Initiative of ETH Zurich. We kindly acknowledge the European Research Council through the Horizon 2020 research and innovation program (grant agreement No. 101002094) for financial support to P.A. Figures 2, 3, 4 and Supplementary Fig. 6 were created with BioRender.com.

## Author contributions

M.G.-G., A.I.B.-M., F.P. and P.A. designed the conceptual framework of the study. M.G-G. performed the experiments. A.I.B-M., M.P., F.S., C.M., K.N., K.M. and L.F. contributed to data acquisition and interpretation. M.G-G. and P.A. wrote the manuscript with contributions from all authors.

## Funding

## Competing interests

The authors declare no competing interests.
