## [Peer Review File · Nature Communications]

Local environment in biomolecular condensates modulates enzymatic activity across length scalesREVIEWER COMMENTS

Reviewer #1 (Remarks to the Author):

The paper by Gil-Garcia et al. describes how the local environment of biomolecular condensates influences the reaction rates of enzymes within the condensate. The authors, through a methodical, careful, and clear study, show that the recruitment of negatively charged cofactors and substrates of an enzyme, NOX, and subsequent increase in reaction rates is significantly correlated with a positively charged condensate microenvironment. The study presents clear and useful controls and does a great job of supporting the key claims with experimental evidence. The findings are interesting – the microenvironment of condensates, rather than simple molecular crowding, dictates reaction rates, and in this case, these increases in reaction rates are condensate size-independent.

There are a few comments below about what can be improved in the paper and what needs to be worked on. In general, however, the paper is excellent.

Introduction

- Line 76-78 – “condensates in cells are typically in the nanometer range”. This claim may not be entirely correct, condensates can often reach micrometer size (e.g. the nucleolus, P-granules, etc). Ref: Forman-Kay et al., RNA 2022.

Results

- Figure 1:

- o E) It's not immediately clear that the FAD cofactor colocalizes with the NADH, and it's also not clear where the condensates are. The green FAD dots don't really line up with the blue NADH blobs, so we don't know which ones are in the same condensate and where the condensates are. A composite image which shows recruitment and localization of NADH and FAD to brightfield condensates would be very helpful.

- Line 167-175: It would be helpful to provide more explanation as to what exactly you're measuring – how much substrate and cofactor is in the condensate compared to the dilute phase? How did you calculate this? Even if the explanation is in methods, it would be nice to have a brief description here. In addition, there should be a sentence describing what the determined partition coefficients actually show – increased positive charge better recruits NADH and FAD?

- Figure 3:

- o C) is unclear. If the protein concentration was close to 300nM, but not at 300nM, why does the schematic show protein concentration at 300nM? What protein concentration was actually used? That should be marked on the graph. Authors should also have some other values on the X axis to give a sense of scale of the binodal. Furthermore, it's unclear what the Y axis represents.

- Line 264-265: Previously, authors said protein concentrations were “close to” 300nM. Here, the authors say the solutions were 300nM. If the saturation concentration is 300nM, wouldn't the proteins undergo phase transition and form condensates? The authors should be more precise in these

paragraphs.

Finally, a recommended experiment— authors can incubate cofactors with only LCD condensates first (concentration required may be different than that for fusions), followed by addition of free NOX. If LCDs are sequestering the substrates (as author's suggest it recruitment of substrates and not NOX), the enzyme activity trend should reverse.

Reviewer #2 (Remarks to the Author):

Gil-Garcia et al. present an interesting work on enzymatic catalysis of NADH oxidase (NOX) in confined microenvironments formed by designed protein condensates. Rather than encapsulating the enzyme NOX in the condensates, as many other have shown before, the authors conjugated NOX to low complexity domains (LCDs) from three different intrinsically disordered proteins, and thus the condensate building block is also the catalyst. This unique approach was previously reported by the authors. In this study, the authors focus on how the chemical composition, and specifically the charge of three different LCDs affect the kinetics of NOX. By using SEC, DLS, and different spectroscopy and microscopy techniques, the authors analyzed the recruitment of conjugated NOX, substrate and cofactor in the condensates and the reaction kinetics in the dilute phase and overall averaged heterogeneous condensate solutions. The most interesting observation of this work is that condensate size does not affect K_{cat} but rather the basic microenvironment created by LLPS of Dpb 1. The mechanism by which the basic environment enhances enzyme catalysis remains unknown, although the authors suggested a few possible explanations. Overall, the paper reads well and presents important insights on how compartmentalization (specifically size and charge of compartments) affects enzymatic catalysis. The thorough analyses and solid data overall support the conclusions of the work, yet I encourage the authors to consider my suggestions as detailed below. I believe that the insights from the work will be of great interest to the growing LLPS community and to the broader readership of Nat. Commun., therefore I recommend on publication of the work after addressing the following comments:

1. Characterization of LLPS: There is little information on the formation of condensates – the authors mention later in the text that the C_{sat} is 300 nM but no phase diagrams are presented to support this. What triggers LLPS? is it salt (20 mM NaCl)? Specifically, the authors stated that “electrostatic interactions are important in the condensation of the selected LCDs”, I’m assuming that the Dbp-NOX will require higher salt concentration to trigger LLPS due to stronger electrostatic repulsion? It will be useful to elaborate on this.
2. Figure 1: (i) Were the microscopy images in panel c of figure 1 taken using phase contrast microscope? If so, please indicate.
(ii) Why did the authors use 5 μ M of conjugated enzyme for the microscopy analysis and a different concentration (1 μ M) to analyze the reaction kinetics?

3. Follow up question: how was the concentration of conjugated enzyme selected? please indicate in Figure 2 caption the protein concentration and conditions used in the reaction kinetics analysis.

4. Page 6 line 149: The line reads as the authors analyzed the recruitment of free NOX rather than the full-length building block (NOX conjugated to LCDs), please rephrase to clarify this point.

5. Have the authors considered analyzing substrate and cofactor partitioning using a complementary method in a similar manner to the enzyme recruitment analysis by using averaged absorbance/fluorescence spectroscopy? This can provide useful information and validation of the confocal analysis.

6. I am curious if the material properties of the difference condensates are different due to their varying net charge. Have the authors analyzed the dynamics of the condensates using FRAP? Is it possible that the diffusion in the condensates affect reaction rate? This is not mandatory but can add some valuable information to the work.

7. The rate of the enzymatic reaction is measured by the decrease in absorbance of the NADH substrate. Have the authors considered the possible contribution of light scattering by the condensates, which might change over time due to droplet coalescence?

8. Do the authors expect that NOX forms dimers also in the dense phase? Does the enzyme form dimers or other type of oligomers in its native form? What is the enzyme pI? Please add this information to the MS. Also, is the enzymatic activity (of free NOX) not compromised by 0.5 M NaCl? Information about the enzyme (pI) might also help explaining the possible interactions with the Dbp 1 LCD as suggested by the authors in the Discussion.

9. The authors state that “for Dbp-NOX, the reaction rate in the dilute phase was lower than the total reaction rate of the two-phase system”. Is it not obvious? How does the fact that the rate in the dilute phase is lower than the dense phase confirms the contribution of the dense phase to the overall enzymatic reaction, considering that the enzyme concentration in the dense phase is much higher? Please remove this statement or rephrase to clarify this point.

10. Are the 100 nm nanoclusters in the dilute phase detectible using electron microscopy? This is not a mandatory analysis but can confirm the DLS results.

11. Figure 4c shows kinetics parameters of 10 nm sized clusters. What is the expected size of the building block (conjugated LCD-NOX)? Could the 10 nm peaks in DLS represent individual building blocks rather than structures?

12. Discussion: “the increase in activity measured in Dbp 1 LCD condensates may indicate that the effect of substrate colocalization becomes important only above a certain threshold”. Please explain this statement.

Reviewer #3 (Remarks to the Author):

The manuscript by Gil-Garcia et al describes a systematic study of the effect of biomolecular condensates on enzymatic rates. This is an important topic that may help us understand both fundamental biological mechanisms and engineering new biocatalysts. The study compares three variants of the enzymes NADH oxidase fused to three different intrinsically disordered regions that drive phase separation. The main finding is that one of these IDRs enhance the enzymatic reaction about threefold, which the authors ascribe to an increase in the apparent k_{cat} . This is in contrast to much of the literature on enzyme enhancement of condensates, which so far has mainly been documented to occur through mass action – i.e. co-partitioning of enzyme and substrate. If the microenvironment truly enhances the fundamental rate constants this will be quite novel and to my mind surprising, but is not entirely supported by the present manuscript. I think this calls for some extra analysis and possibly simple extra experiment to clarify whether the rate enhancement is due to up concentration of (co-)substrate or not. The manuscript itself also seems to be somewhat in doubt about this, as the enhancement is sometimes attributed to k_{cat} and sometimes to co-partitioning.

Specific comments:

1. The key question in my mind is to clarify whether the rate enhancement is due to substrate co-partitioning or not. Two out of the three substrates partition into the condensate with the enzyme. All other factors being equal, this suggests that the rate should be enhanced by mass action. When no enhancement is observed for two out of three this suggests a “cancellation of effects”, where rate enhancement due to mass action is cancelled by e.g. lower k_{cat} or mass transport limitations. A good starting point would be to decompose the rates into the contribution from the two phases theoretically. The authors do the initial steps towards doing this by defining rates r_I and r_{II} on line 241. I think this analysis should be carried to the end to see what the actual rates are in the dense phase and compare them to predictions from simple mass action.
2. An alternative explanation to the rate enhancement is the elevated concentration of the co-substrate in the condensate. To rule out this effect, it would be nice to see the dependence of the rates on the FAD concentration.
3. Fig. 4 – Michaelis-Menten dependence – The Michaelis-Menten curve and the simple effect via the apparent k_{cat} is surprising. A similar K_M means that the true K_M of the dense phase has to be 12.5-fold greater (as the concentration of NADH is this much greater. Again, not impossible but a remarkable coincidence that should be discussed. I also think the non-Michaelis-Menten behavior of the two other enzymes (see below) call for a wider range of concentrations to be tested.
4. L. 189 “showing similar relative trends between the different systems” – Is this really true? Two of the enzymes show a decrease at high concentrations of NADH, which suggests a non Michaelis-Menten mechanism. This is glossed over and somewhat mis-represented by the statement above. There are good reasons why the rate may drop, but it should be discussed openly and included in the main manuscript.
5. Oxygen is also a substrate and could in principle also contribute to the rate enhancement. It would be very difficult to measure partitioning of oxygen, and small molecules do not have particularly large partitioning constants, so I think this is unlikely. However, it would perhaps be worth mentioning in passing, and for example state if it is saturated with oxygen under the conditions used.
6. The kinetics of Dbp1-NOX in Fig 2A are clearly multi-phasic in contrast to the other proteins. This is the protein that the entire story hinges on, so I think we need an explanation for this. The initial fast burst

phase is really what drives the difference between studies. Is there something fundamentally different about this reaction that make it multiphasic?

Minor issues:

1. Could the sequences in the SI be added as text rather than images to make them machine-readable and easier to copy?
2. 2. L. 171 – You cannot conclude that they scale exponentially from 3 data points.
3. Fig 1F – Error bars missing

Reviewer #1 (Remarks to the Author):

The paper by Gil-Garcia et al. describes how the local environment of biomolecular condensates influences the reaction rates of enzymes within the condensate. The authors, through a methodical, careful, and clear study, show that the recruitment of negatively charged cofactors and substrates of an enzyme, NOX, and subsequent increase in reaction rates is significantly correlated with a positively charged condensate microenvironment. The study presents clear and useful controls and does a great job of supporting the key claims with experimental evidence. The findings are interesting – the microenvironment of condensates, rather than simple molecular crowding, dictates reaction rates, and in this case, these increases in reaction rates are condensate size-independent.

There are a few comments below about what can be improved in the paper and what needs to be worked on. In general, however, the paper is excellent.

Answer from the authors: We thank the reviewer for the overall positive feedback and the constructive comments, which we have addressed as described in the point-to-point responses below.

Introduction

• Line 76-78 – “condensates in cells are typically in the nanometer range”. This claim may not be entirely correct, condensates can often reach micrometer size (e.g. the nucleolus, P-granules, etc). Ref: Forman-Kay et al., RNA 2022.

Answer from the authors: We thank the reviewer for the suggestion. We have removed the word “typically” and rephrased accordingly in the new version of the manuscript.

“Moreover, condensates in cells can not only populate the microscale, but also the nanoscale³⁰”

Results

• Figure 1:

o E) It's not immediately clear that the FAD cofactor colocalizes with the NADH, and it's also not clear where the condensates are. The green FAD dots don't really line up with the blue NADH blobs, so we don't know which ones are in the same condensate and where the condensates are.

Answer from the authors: We appreciate the reviewer's observation. Due to the short timescale of the reaction, to evaluate the partition coefficient we had to conduct two distinct experiments, adding substrate and cofactor individually. This allows to avoid interference of the fast reaction while measuring the partitioning coefficient. We have now specified in the caption of Figure 1 that the experiments were conducted independently and further clarified this point in the main text (Results section “Condensate microreactors with different local environments”).

Figure: “**Figure 1E:** Representative fluorescence confocal microscopy images showing the recruitment of NADH (top, blue fluorescence) and FAD (bottom, green fluorescence) in condensates of Ddx4-NOX (left), Laf1-NOX (middle) and Dbp1-NOX (right). NADH and FAD were added individually in two distinct experiments to avoid interference of the rapid reaction. Scale bar represents 5 μm ”

Results section: “To avoid any influence of the rapid reaction, we measured the partitioning of NADH and FAD independently by confocal microscopy. Specifically, we recorded their intrinsic fluorescence signal at 410-440 nm and 510-530 nm for NADH and FAD, respectively (Figure 1E), and calculated the partition coefficient (K_p) as the ratio of fluorescence intensity inside and outside the condensates.”

A composite image which shows recruitment and localization of NADH and FAD to brightfield condensates would be very helpful.

Answer from the authors: Prompted by the comment of the reviewer, we performed a co-localization experiment adding brightfield images, again by adding the two molecules individually in two different samples to avoid interference of the reaction (new **Supplementary Figure 3**). The blue and green fluorescence signals of NADH and FAD overlap with the condensates observed in the bright-field images.

Supplementary Figure 3. Substrate and cofactor partition into Dbp1-NOX condensates. Representative fluorescence and bright-field confocal microscopy images showing the recruitment of NADH alone (top, blue fluorescence) and FAD alone (bottom, green fluorescence) in Dbp1-NOX condensates. The merged images confirm the localization of substrate and cofactor within the condensates.

• Line 167-175: It would be helpful to provide more explanation as to what exactly you're measuring – how much substrate and cofactor is in the condensate compared to the dilute phase? How did you calculate this? Even if the explanation is in methods, it would be nice to have a brief description here. In addition, there should be a sentence describing what the determined partition coefficients actually show – increased positive charge better recruits NADH and FAD?

Answer from the authors: We have now specified this information in the Results section “Condensate microreactors with different local environments” of the main text, and added the suggested sentence.

“Specifically, we recorded their intrinsic fluorescence signal at 410-440 nm and 510-530 nm for NADH and FAD, respectively (Figure 1E), and calculated the partition coefficient (K_p) as the ratio of fluorescence intensity inside and outside the condensates.”

“Both molecules specifically partition into the condensates, as confirmed by the overlap between the fluorescence and bright-field images for Dbp1-NOX condensates (similar results were obtained for the other LCD-NOX constructs) (Suppl. Fig. 3). These findings confirm the increased uptake of the negatively-charged substrate and cofactor by the most cationic condensates.”

• Figure 3:

o C) is unclear. If the protein concentration was close to 300nM, but not at 300nM, why does the schematic show protein concentration at 300nM? What protein concentration was actually used? That should be marked on the graph.

Answer from the authors: The C_{sat} was estimated from the measurement of the concentration in the dilute phase after separation of the dense phase, and was equal to 290-300 nM (Table 1). This value has been now further confirmed by new experiments on phase diagrams performed to address point 1 of reviewer 2 (new **Supplementary Figure 7** and **Supplementary Figure 8**). We performed experiments at 280 nM to be close to this value. At this protein concentration we see only a small amount of very tiny condensates. We have now specified that experiments were performed at 280 nM in both the figure and the text.

Authors should also have some other values on the X axis to give a sense of scale of the binodal. Furthermore, it's unclear what the Y axis represents.

We have added more values on the X axis as requested (see new Figure 3C below). Y axis represents temperature normalized by the interaction coefficient (χ). This has been now specified in the Figure caption.

Figure 3C: Schematic phase diagram showing the protein concentration at which nanoclusters were formed (280 nM, red square), and the c_{sat} (300 nM). The Y-axis represents temperature normalized by the interaction coefficient (x).

• Line 264-265: Previously, authors said protein concentrations were “close to” 300nM. Here, the authors say the solutions were 300nM. If the saturation concentration is 300nM, wouldn’t the proteins undergo phase transition and form condensates? The authors should be more precise in these paragraphs.

Answer from the authors: Following the reviewer’s comment, we have now specified that the working concentration was 280 nM in the different sections and figures throughout the manuscript.

Finally, a recommended experiment— authors can incubate cofactors with only LCD condensates first (concentration required may be different than that for fusions), followed by addition of free NOX. If LCDs are sequestering the substrates (as author's suggest it recruitment of substrates and not NOX), the enzyme activity trend should reverse.

Answer from the authors: We thank the reviewer for this interesting suggestion. Unfortunately, the very low total volume fraction of the dense phase prevents this experiment, since an insufficient amount of substrate is depleted from the dilute phase to measure a detectable difference in the reaction in the dilute phase in this experiment. Specifically, considering the total volume fraction of 0.015%, even with a high partition coefficient of 50, less than 1% of the substrate would be sequestered into the LCD condensates. Moreover, we have observed that free NOX partially partitions into LCD condensates.

Reviewer #2 (Remarks to the Author):

Gil-Garcia et al. present an interesting work on enzymatic catalysis of NADH oxidase (NOX) in confined microenvironments formed by designed protein condensates. Rather than encapsulating the enzyme NOX in the condensates, as many other have shown before, the authors conjugated NOX to low complexity domains (LCDs) from three different intrinsically disordered proteins, and thus the condensate building block is also the catalyst. This unique approach was previously reported by the authors. In this study, the authors focus on how the chemical composition, and specifically the charge of three different LCDs affect the kinetics of NOX. By using SEC, DLS, and different spectroscopy and microscopy techniques, the authors analyzed the recruitment of conjugated NOX, substrate and cofactor in the condensates and the reaction kinetics in the dilute phase and overall averaged heterogeneous condensate solutions. The most interesting observation of this work is that condensate size does not affect K_{cat} but rather the basic microenvironment created by LLPS of Dpb 1. The mechanism by which the basic environment enhances enzyme catalysis remains unknown, although the authors suggested a few possible explanations. Overall, the paper reads well and presents important insights on how compartmentalization (specifically size and charge of compartments) affects enzymatic catalysis. The thorough analyses and solid data overall support the conclusions of the work, yet I encourage the authors to consider my suggestions as detailed below. I believe that the insights from the work will be of great interest to the growing LLPS community and to the broader readership of Nat. Commun., therefore I recommend on publication of the work after addressing the following comments:

Answer from the authors: We thank the reviewer for the positive feedback and the constructive comments, which we have addressed as described in the point-to-point responses below.

1. Characterization of LLPS: There is little information on the formation of condensates – the authors mention later in the text that the C_{sat} is 300 nM but no phase diagrams are presented to support this.

Answer from the authors: The c_{sat} was estimated by measuring the monomer concentration in the dilute phase via size exclusion chromatography coupled with intrinsic fluorescence detection (Table 1). Following the reviewer suggestion, we have now measured the phase diagram of the three proteins at different protein concentrations and constant ionic strength (20 mM NaCl), by analyzing the presence of microscopic condensates via bright-field microscopy. These phase diagrams are now included in the new version of the manuscript (**Supplementary Figure 7**) and corroborate the presence of micron-sized condensates at concentrations higher than 300 nM. A new **Supplementary Figure 8** showing representative bright-field microscopy images of samples at different protein concentration and constant ionic strength has been added to the manuscript.

Supplementary Figure 7. Phase diagram of the different LCD-NOX fusion proteins at different protein concentration and constant ionic strength of 20 mM NaCl. Green circles and red cross indicate presence and absence of phase separation, respectively.

Supplementary Figure 8. Representative bright-field microscopy images of LCD-NOX samples at different protein concentration and constant ionic strength of 20 mM NaCl. Scale bar represents 50 μm .

What triggers LLPS? is it salt (20 mM NaCl)?

The trigger of phase separation for these three LCDs is the decrease in salt concentration, as previously described in the literature (Laf1: Elbaum-Garfinkle S, et al., *The disordered P granule protein LAF-1 drives phase separation into droplets with tunable viscosity and dynamics. Proc Natl Acad Sci U S A. 2015*, Dbp1: Faltova L, et al., *Multifunctional Protein Materials and Microreactors using Low Complexity Domains as Molecular Adhesives. ACS Nano. 2018*, Ddx4: Nott TJ, et al., *Phase transition of a disordered nuage protein generates environmentally responsive membraneless organelles. Mol Cell. 2015*). As shown in the phase diagram in Figure 3F, LCD-NOX condensates only form at low salt concentration (below 300 mM NaCl), due to the important role played by electrostatic interactions and cation- π contacts in the self-assembly of the three LCDs. We have included the 3 selected references in the new version of the manuscript (Results section “Condensate microreactors with different local environments”).

“Electrostatic interactions are important in the condensation of the selected LCDs, and the phase separation of these sequences can be modulated by salt concentration^{35,43,44}.”

Specifically, the authors stated that “electrostatic interactions are important in the condensation of the selected LCDs”, I’m assuming that the Dbp-NOX will require higher salt concentration to trigger LLPS due to stronger electrostatic repulsion? It will be useful to elaborate on this.

Although the dependence of salt concentration indicates an important role of electrostatics, these LCDs mediate a variety of intermolecular interactions, not only electrostatics. Moreover, not only the net charge, but also the number and type of sticker and spacer residues influence the phase separation process. All these effects could compensate the difference in net charge and lead to similar critical salt concentration required for phase separation.

2. Figure 1: (i) Were the microscopy images in panel c of figure 1 taken using phase contrast microscope? If so, please indicate.(ii) Why did the authors use 5 μ M of conjugated enzyme for the microscopy analysis and a different concentration (1 μ M) to analyze the reaction kinetics?

Answer from the authors: (i) Yes, the images in Figure1C were taken using phase contrast microscopy. This information has been now added into the manuscript.

“Representative phase-contrast microscopy images of solutions of 5 μ M Ddx4-NOX (top), Dbp1-NOX (bottom, left) and Laf1-NOX (bottom, right) in 25 mM Tris, 20 mM NaCl, pH 7.5 showing the presence of micron-sized droplets when the enzyme is fused to LCDs. Scale bar represents 5 μ m”

(ii) the differences in protein concentration for both experiments stem from technical reasons. We worked at 5 μ M to ensure the formation of sufficiently large condensates, facilitating the determination of the partition coefficient of the substrate and cofactor. However, at this protein concentration the reaction is too fast to be monitored. For this reason, we selected a lower protein concentration of 1 μ M, which allowed us to track the reaction over time. Importantly, this lower concentration still enabled the formation of condensates by all three LCD-NOX constructs (Supplementary Figure 8).

3. Follow up question: how was the concentration of conjugated enzyme selected? please indicate in Figure 2 caption the protein concentration and conditions used in the reaction kinetics analysis.

Answer from the authors: We selected the concentration of the conjugated enzyme based on the enzymatic reaction rate. We chose a protein concentration that not only facilitated the formation of protein condensates for all three LCD-NOX constructs but also allowed for monitoring of the reaction over a suitable time scale.

The information corresponding to the protein concentration and conditions used in the enzymatic kinetics has been now added to the Figure 2 caption.

“Representative profile of the reaction progress characterized by a decrease in the NADH absorbance at 340 nm for homogeneous solution (black symbols, NOX) and the heterogeneous system composed of droplets and the dilute phase (blue symbols, Dbp1-NOX, red symbols, Laf1-NOX and green symbols, Ddx4-NOX). Proteins were diluted to 1 μ M in 25 mM Tris, 20 mM NaCl pH 7.5”

4. Page 6 line 149: The line reads as the authors analyzed the recruitment of free NOX rather than the full-length building block (NOX conjugated to LCDs), please rephrase to clarify this point.

Answer from the authors: Thank you for pointing this out. We have rephrased this sentence in the new version of the manuscript.

“We next determined the recruitment of the different LCD-NOX fusion proteins into the different condensates by separating the dilute and dense phase by centrifugation and measuring the protein concentration in the dilute phase by size exclusion chromatography (SEC).”

5. Have the authors considered analyzing substrate and cofactor partitioning using a complementary method in a similar manner to the enzyme recruitment analysis by using averaged absorbance/fluorescence spectroscopy? This can provide useful information and validation of the confocal analysis.

Answer from the authors: We thank to the reviewer for this interesting suggestion. We tried to separate dilute and dense phase and measure remaining concentration of substrate and cofactor in the dilute phase, comparing with a control homogeneous solution. However, this was not feasible due to the very low total volume of the dense phase and the minor changes in the concentration of the substrate in the dilute phase upon recruitment in the dense phase. Specifically, if we consider the total volume fraction of 0.015%, even with a high partition coefficient of 50, less than 1% of the substrate amount would be sequestered into the condensates, making very difficult to observe any change in the absorbance/fluorescence of these molecules via spectroscopy.

6. I am curious if the material properties of the difference condensates are different due to their varying net charge. Have the authors analyzed the dynamics of the condensates using FRAP? Is it possible that the diffusion in the condensates affect reaction rate? This is not mandatory but can add some valuable information to the work.

Answer from the authors: The reviewer raised a very interesting point. However, this is a complex problem that requires a dedicated work and is out of the scope of this paper.

7. The rate of the enzymatic reaction is measured by the decrease in absorbance of the NADH substrate. Have the authors considered the possible contribution of light scattering by the condensates, which might change over time due to droplet coalescence?

Answer from the authors: We appreciate the reviewer’s comment. To address this concern, we performed a time course experiment in which we recorded Abs_{340nm} of NADH in the presence of droplets of Dbp1-NOX and in the absence of any enzymatic reaction (we did not add the cofactor (FAD)). The goal was to determine whether, as suggested by the reviewer, the NADH absorbance values were affected by the condensate coalescence process over a 30-min period. This timescale is considerably longer than the time required for the enzymatic reaction (which occurs within minutes). As illustrated in the **new Supplementary Figure 4**, the absorbance values of NADH were unaffected by the light scattering of droplets. We specify this information in the revised version of the manuscript (Results section “Condensation with suitable local environment modulates NOX enzymatic activity”).

“To confirm the absence of any potential effect of the condensates on the absorbance signal at 340 nm, we monitored the absorbance of NADH in the presence of Dbp1-NOX condensates in the absence of reaction (i.e., in the absence of FAD cofactor), observing a negligible change of the signal over time (Suppl. Fig. 4).”

Supplementary Figure 4. Measurement of NADH concentration in the presence of Dbp1-NOX condensates in the absence of cofactor (no reaction occurring). Representative profile of the NADH concentration measured by absorbance at 340 nm during 30 min in the presence of 1 μ M Dbp1-NOX condensates without cofactor.

8. Do the authors expect that NOX forms dimers also in the dense phase? Does the enzyme form dimers or other type of oligomers in its native form?

Answer from the authors: According to our mass photometry data acquired at 20 nM NOX (Figure 2D), NOX protein assembles as a homodimer and the appended LCD does not affect the ability of NOX to dimerize. The formation of dimers in native conditions was also demonstrated in the following paper: *Hecht HJ, et al., Crystal structure of NADH oxidase from Thermus thermophilus. Nat Struct Biol. 1995.* It is difficult to characterize the oligomeric state in the dense phase. Given the high concentrations of LCD-NOX in the condensates (in the mM range), we would expect NOX to form oligomers also in the dense phase.

What is the enzyme pI? Please add this information to the MS.

The pI is 8.86 (calculated using ProtParam: <https://web.expasy.org/protparam/>) and this information has been now included in the manuscript (Introduction section).

“Moreover, the enzyme presents an isoelectric point of 8.86 and assembles in dimers, therefore further promoting multivalent interactions and condensation.”

Also, is the enzymatic activity (of free NOX) not compromised by 0.5 M NaCl?

The activity of free NOX at 0.5 M NaCl is not compromised, as observed in the experiments reported in Figure 2C.

Information about the enzyme (pI) might also help explaining the possible interactions with the Dbp 1 LCD as suggested by the authors in the Discussion.

Prompted by the reviewer’s suggestion, we created the NOX dimeric structure from PDB 1NOX using PISA (https://www.ebi.ac.uk/msd-srv/prot_int/cgi-bin/piserver) and analyzed the surface electrostatics using PyMOL and the incorporated APBS electrostatics plugin. The pI indicates that the enzyme should be positively charged at physiological pH (net charge +2).

This pI can be a good indicative when the protein does not have a 3D structure and all the residues are solvent exposed, however, since this protein is folded and forms homodimers, the pI is not the most reliable parameter to consider. Consequently, we analyzed the surface charge distribution of the dimeric structure. A detailed examination of the surface electrostatics revealed two partially acidic regions (indicated in the attached Supplementary Figure 13) that can potentially establish electrostatic contacts with the cationic Dbp1 LCD. This might be a putative explanation about the contacts between the LCD and the NOX and we have included this hypothesis in the “Discussion and conclusions” section.

“A possible reason behind this change is the presence of specific interactions between the Dbp1 LCD and the NOX enzyme which occur only in the dense phase due to conformational changes in the condensates with respect to the dilute solution. Indeed, dimeric NOX presents two anionic patches on the surface that could potentially interact with the cationic Dbp1 LCD via electrostatic interactions (Suppl. Fig. 13).”

Supplementary Figure 13. Surface electrostatic representations of dimeric NOX (PDB 1NOX). Residues were colored according to their anionic (red) or cationic (blue) character using the APBS electrostatics plugin in PyMOL. The anionic patches are indicated by a black square.

9. The authors state that “for Dbp-NOX, the reaction rate in the dilute phase was lower than the total reaction rate of the two-phase system”. Is it not obvious? How does the fact that the rate in the dilute

phase is lower than the dense phase confirms the contribution of the dense phase to the overall enzymatic reaction, considering that the enzyme concentration in the dense phase is much higher? Please remove this statement or rephrase to clarify this point.

Answer from the authors: We have rephrased the sentence to clarify this point. The recruitment of enzyme in the dense phase leads to a lower rate in the dilute phase compared to the total rate. However, the decrease in the dilute phase is different for Dbp1-NOX compared to the other two constructs, as discussed in the paragraph in the main text that follows this sentence. Moreover, the comparison between the rate of the dilute phase and the rate of the two-phase system contains important information on the reaction locus. Although it is true that a significant amount of enzyme is incorporated in the condensates, the environment in the condensates could in principle promote a decrease of the reaction rate due to different factors such as mass transfer limitations and decrease in the activity coefficient of the enzyme, which could compensate the high local concentration. Moreover, in our system we observed an increase of the reaction rate in the dilute phase compared to the homogeneous solution, likely due to the presence of nanoclusters. This further shows the different possible scenarios that can occur with phase-separating system. Therefore, it is important to analyze the relative contribution of the dilute and dense phase to the overall rate, which is not obvious a priori.

“In analogy with Laf1- and Ddx4-NOX, also for Dbp1-NOX the reaction rate in the dilute phase is lower than the total reaction rate of the two-phase system due to the recruitment of the enzyme in the dense phase”

10. Are the 100 nm nanoclusters in the dilute phase detectible using electron microscopy? This is not a mandatory analysis but can confirm the DLS results.

Answer from the authors: Following the suggestion of the reviewer we made several attempts to detect the nanoclusters in the dilute phase using transmission electron microscopy by negative staining. However, it was very difficult to discern between real clusters or artifacts. Moreover, since the clusters are very sensitive to the buffer conditions, we cannot discard any potential effect (e.g. cluster dissolution) of the washing steps and the staining process using uranyl acetate salt. Also, their small volume fraction could difficult their identification using TEM.

As a more quantitative orthogonal method, we applied Nanoparticle Tracking Analysis (NTA) following the same approach used for the analysis of samples below the C_{sat} (Figure 3E). As shown in the new **Supplementary Figure 6**, the dilute phase of the three LCD-NOX proteins also contains clusters with an average diameter of 100-150 nm, confirming the DLS data. NTA also provides with images of the clusters. (see Figures below).

Supplementary Figure 6. Size distribution of the dilute phase after removal of the micron-sized condensates by centrifugation measured by nanoparticles tracking analysis (NTA). The results confirm the presence of nanoclusters.

Representative image of Dbp1-NOX clusters in the dilute phase measured by NTA.

11. Figure 4c shows kinetics parameters of 10 nm sized clusters. What is the expected size of the building block (conjugated LCD-NOX)? Could the 10 nm peaks in DLS represent individual building blocks rather than structures?

Answer from the authors: We thank the reviewer for giving us the opportunity to clarify this point. With 10 nm we indicated the estimated size scale of the dimer, corresponding to the state of NOX in homogeneous solutions (without LCDs) (no clusters). We estimated a more accurate value of the size scale of the dimers of NOX/LCD-NOX by modeling the 3D structure of the protein using AlphaFold2 and the software HullRad (*P. J. Fleming & K. G. Fleming. HullRad: Fast Calculations of Folded and Disordered Protein and Nucleic Acid Hydrodynamic Properties. Biophys J., 2018*) obtaining a hydrodynamic diameter of the dimer of 5.7 and 8.2 nm, respectively. We have specified what symbolizes each size in the new Figure 4 cation.

“Figure 4D) Schematic illustration showing the effect of protein condensation on the normalized rates across length scales. The size of 10 nm, 100 nm and 1 μ m represents the dimeric version of the protein, the nanoclusters and the condensates, respectively.”

12. Discussion: “the increase in activity measured in Dbp 1 LCD condensates may indicate that the effect of substrate colocalization becomes important only above a certain threshold”. Please explain this statement.

Answer from the authors: We appreciate the reviewer’s comment. Moreover, new experiments performed to address a comment of reviewer 3 (new **Supplementary Figure 11**) excludes a unique role of the cofactor. We have now removed this sentence in the new version of the manuscript.

Reviewer #3 (Remarks to the Author):

The manuscript by Gil-Garcia et al describes a systematic study of the effect of biomolecular condensates on enzymatic rates. This is an important topic that may help us understand both fundamental biological mechanisms and engineering new biocatalysts. The study compares three variants of the enzymes NADH oxidase fused to three different intrinsically disordered regions that drive phase separation. The main finding is that one of these IDRs enhance the enzymatic reaction about threefold, which the authors ascribe to an increase in the apparent k_{cat} . This is in contrast to much of the literature on enzyme enhancement of condensates, which so far has mainly been documented to occur through mass action – i.e. co-partitioning of enzyme and substrate. If the microenvironment truly enhances the fundamental rate constants this will be quite novel and to my mind surprising, but is not entirely supported by the present manuscript. I think this calls for some extra analysis and possibly simple extra experiment to clarify whether the rate enhancement is due to up concentration of (co-)substrate or not. The manuscript itself also seems to be somewhat in doubt about this, as the enhancement is sometimes attributed to k_{cat} and sometimes to co-partitioning.

Answer from the authors: We thank the reviewer for the positive feedback and the constructive comments. We have performed additional modeling analysis and extra experiment to clarify this important point. See detailed answers below. In short, the integration of experimental data, model analysis and literature findings robustly indicate a rate enhancement resulting from a change in k_{cat} within the dense phase, rather than being attributed to mass action.

Specific comments:

1. The key question in my mind is to clarify whether the rate enhancement is due to substrate co-partitioning or not. Two out of the three substrates partition into the condensate with the enzyme. All other factors being equal, this suggests that the rate should be enhanced by mass action. When no enhancement is observed for two out of three this suggests a “cancellation of effects”, where rate enhancement due to mass action is cancelled by e.g. lower k_{cat} or mass transport limitations. A good starting point would be to decompose the rates into the contribution from the two phases theoretically. The authors do the initial steps towards doing this by defining rates r_l and r_{ll} on line 241. I think this analysis should be carried to the end to see what the actual rates are in the dense phase and compare them to predictions from simple mass action.

Answer from the authors: We thank the reviewer for giving us the opportunity to clarify this important point. We first note that the increase in substrate concentration in the dense phase does not necessarily lead to an increase in the average reaction rate of the entire system (dilute + dense phase) due to mass action. This is due to the fact that reaction rates depend on the activities, and the activity coefficient of the substrate in the dense phase can be much smaller than the one in the dilute phase. A recent work illustrates how the decrease of the activity coefficient could exactly compensate the increase in concentration (Bauermann J, Laha S, McCall PM, Jülicher F, Weber CA. *Chemical Kinetics and Mass Action in Coexisting Phases. J Am Chem Soc. 2022*). In this situation, the average rate of the system (dilute + dense phase) can be different from the homogeneous solution only if k_{cat} or K_M are different in the dense and dilute phase.

The absence of mass action effect is also shown by the absence of enhancement observed in two out of three systems (Laf1-NOX and Ddx4-NOX), despite they all locally concentrate substrates. I.e. the changes in enzymatic rate are not proportional to the partitioning of the substrates. The “cancellation of effects” mentioned by the reviewer is theoretically possible. However, the two systems (Laf1-NOX and Ddx4-NOX), which exhibit different partitioning of substrates ($K_p = 5.8 \pm 0.7$, and 2.2 ± 0.3), should have different cancellations of effects that result in exactly the same rate, which is also equal to the rate of the homogeneous system (rate Laf1-NOX = rate Ddx4-NOX = rate homogeneous system). This scenario is extremely unlikely.

Moreover, as discussed below in the answer to point 2, the initial rates of the Dbp1-NOX heterogeneous system reach a plateau value at approximately 200 μM FAD. This indicates that the effect of up-concentration of cofactor cannot be the reason of the observed increment in the reaction rates for this particular system.

Altogether, the results point to an increase in k_{cat} in the dense phase of Dbp1-NOX.

Following the suggestion of the reviewer, we have decomposed the rates into the contribution of the different phases. This new analysis reveals a dependence of the rates on the size of the condensates. We measured the diffusivity of small molecules in the dense phase, and these new data indicate possible presence of mass transport limitations, which limit the estimation of the true k_{cat} in the dense phase by model analysis.

We have significantly expanded the main text to clarify this crucial point (Results section “Modulation of enzymatic activity by local environment occurs across different length scales and is larger in nanoclusters”):

“The results show that the change in the overall enzymatic activity due to the local microenvironment of the Dbp1-NOX dense phase occurs across length scales.

We further analyzed the enhancement of the reaction rate in the dense phase by deconvoluting the contribution of the individual phases to the average reaction rate. We started from the 280 nM solution, where nanoclusters are present. The average rate can be expressed as $r = r_1\Phi_1 + r_2\Phi_2$, where the indexes 1, and 2 indicate, respectively, the dimers in the dilute phase and the nanoclusters. Φ represents the volume fractions, which have been previously estimated experimentally. Assuming the same rate in the dilute and homogenous phase and normalizing for the enzyme concentration in the two phases (see details in Materials and Methods), we estimated $r_{2, \text{norm}} = 6.1 \cdot r_{1, \text{norm}}$. Such increment could be due to either a mass action effect related to the partitioning of substrate and cofactors, or to a change in the kinetic parameters. Recent theoretical work highlights that the activity coefficient of client molecules in the dense phase can be significantly reduced compared to the dilute phase and can compensate for the local increase in concentration⁵⁰. Changes of the total reaction rate (of both dilute and dense phase) compared to the homogeneous solution should therefore be attributed to changes in the kinetic parameters of the enzyme rather than mass action laws⁵⁰. The absence of mass action effect in our

systems is also supported by the absence of enhancement observed in two out of three systems (Laf1-NOX and Ddx4-NOX), despite they all locally concentrate substrates with different partitioning coefficients (**Figure 1F**), i.e. the changes in the reaction rate are not proportional to the partitioning of substrate and cofactor. Moreover, measurements of the initial rates for the Dbp1-NOX heterogeneous system at different cofactor concentrations showed a nonlinear increase of activity with cofactor concentration, reaching a plateau at 200 μM (**Suppl. Fig. 11**).

All together, these observations point to an increase in the k_{cat} in the dense phase as the reason behind the rate enhancement in the Dbp1-NOX system.

We note that in our system the enzyme is incorporated in the scaffold protein of the condensates and does not participate in the interactions driving phase separations, which are largely mediated by the LCDs. A key feature of our strategy is that the change in the activity coefficient of the enzyme in the dense phase is therefore negligible, and high enzymatic rates can be achieved inside the dense phase. The enzymatic reaction is always locally accelerated in the condensates compared to the surrounding dilute phase due to the larger concentration of the enzyme in the dense phase, even for the constructs that do not show a rate enhancement (**Figure 4C**).

Finally, we evaluated the effect of the size of the condensates on the increase in enzymatic rate by estimating the contribution of the micron-sized condensates and the nanoclusters for the 1 μM Dbp1-NOX sample. The average reaction rate can be expressed as $r = r_1\Phi_1 + r_2\Phi_2 + r_3\Phi_3$, where the indexes 1, 2 and 3 indicate, respectively, the dimers in the dilute phase, the nanoclusters and the micron-sized condensates. Assuming the same rate in the nanoclusters in the solutions at 280 nM and 1 μM protein, and normalizing again for the enzyme concentration (see Materials and Methods) we obtained $r_{3,\text{norm}} = 2.1 \cdot r_{1,\text{norm}}$ which indicates a larger rate for the nanoclusters ($r_2 = 2.9 \cdot r_3$) compared to the micron-sized condensates. This result can be due to the presence of mass transfer limitations. To test this hypothesis, we estimated the characteristic diffusion time of a small molecule in the condensates by fluorescence correlation spectroscopy (FCS) (**Suppl. Fig. 12**). Considering a characteristic radius L of 1 μm for the condensates, based on simple scaling analysis the measured diffusion coefficient D of 3.8 $\mu\text{m}^2/\text{s}$ corresponds to a characteristic diffusion time of $T_D = L^2/D$ of 263 ms. This time is comparable to the estimated characteristic time of the reaction $T_R = 1/k_{\text{cat}}$ of 258 ms (the Damköhler number, $Da = T_D/T_R$, is approximately 1), indicating possible presence of mass transfer limitations. However, we cannot exclude that the larger rate for smaller condensates is potentially due to also a higher reactivity of the enzyme at the interface of the condensates.”

To further clarify the different rates measured in the manuscript, we also have included a schematic illustration in Figure 4C).

Figure 4C. Schematic illustration of the initial rates in the dilute and dense phase (blue color) and of the overall initial rate of the entire system (red color) for the different systems (NOX homogeneous system and LCD-NOX heterogeneous systems).

2. An alternative explanation to the rate enhancement is the elevated concentration of the co-substrate in the condensate. To rule out this effect, it would be nice to see the dependence of the rates on the FAD concentration.

Answer from the authors: Following the suggestion of the reviewer we have analyzed the initial rates of Dbp1-NOX heterogeneous system at different cofactor (FAD) concentrations. As shown in the **new Supplementary Figure 11**, the initial rates of the Dbp1-NOX heterogeneous system reach a plateau value at approximately 200 μM FAD. This indicates that the effect of up-concentration of cofactor cannot be the reason of the observed increment in the reaction rates for this particular system.

Supplementary Figure 11. Dependence of Dbp1-NOX initial rates on cofactor concentration. Initial rates of the Dbp1-NOX heterogeneous system measured by NADH absorbance at 340 nm at various FAD concentrations.

We have added the following paragraph in the Results section “Enhancement of enzymatic rate by local environment occurs across different length scales and is larger in nanoclusters”:

“Moreover, measurements of the initial rates for the Dbp1-NOX heterogeneous system at different cofactor concentrations showed a nonlinear increase of activity with cofactor concentration, reaching a plateau at 200 μM (Suppl. Fig. 11).”

3. Fig. 4 – Michaelis-Menten dependence – The Michaelis-Menten curve and the simple effect via the apparent k_{cat} is surprising. A similar K_M means that the true K_M of the dense phase has to be 12.5-fold greater (as the concentration of NADH is this much greater. Again, not impossible but a remarkable coincidence that should be discussed. I also think the non-Michaelis-Menten behavior of the two other enzymes (see below) call for a wider range of concentrations to be tested.

Answer from the authors: Connected to the answer to point 1, the activity coefficient of the substrate can be much lower than in the dilute phase. A recent work (*Bauermann J, et al.. Chemical Kinetics and Mass Action in Coexisting Phases. J Am Chem Soc. 2022*) illustrates how the decrease of the activity coefficient can actually match the partition coefficient, leading to essentially the same K_M in the two phases.

Following the reviewer’s suggestion, we have tested the initial rates at higher substrate concentrations (300 μM and 500 μM) (see new **Supplementary Figure 10**) for the 4 different systems, observing a decrease in the initial rates for all systems.

Supplementary Figure 10. Condensates composed of different chimeric proteins alter NOX enzymatic activity. Initial rates of the heterogeneous systems (presence of condensates) and the NOX homogeneous system at different substrate concentrations.

4. L. 189 “showing similar relative trends between the different systems” – Is this really true? Two of the enzymes show a decrease at high concentrations of NADH, which suggests a non Michaelis-Menten mechanism. This is glossed over and somewhat mis-represented by the statement above. There are good reasons why the rate may drop, but it should be discussed openly and included in the main manuscript.

Answer from the authors: As illustrated in the new **Supplementary Figure 10**, the Dbp1-NOX

heterogeneous system exhibits substrate inhibition at high NADH concentrations. Therefore, we applied the Michaelis-Menten equation in the concentration range between 50 μM and 200 μM , where the enzyme shows a Michaelis-Menten behavior, to determine the apparent K_M and k_{cat} .

We have added sentences to openly discuss this in the main manuscript:

“In the range of substrate concentrations from 50 to 200 μM , the data could be described by a simple Michaelis-Menten kinetics (Figure 4). An inhibitory effect of the substrate and a more complex behavior was observed at higher NADH concentrations (Suppl. Fig. 10), which were therefore not considered for the analysis.”

5. Oxygen is also a substrate and could in principle also contribute to the rate enhancement. It would be very difficult to measure partitioning of oxygen, and small molecules do not have particularly large partitioning constants, so I think this is unlikely. However, it would perhaps be worth mentioning in passing, and for example state if it is saturated with oxygen under the conditions used.

Answer from the authors: We thank the reviewer for the important suggestion. We have included a comment about the potential role of oxygen in the rate enhancement in the “Conclusions and discussion” section of the manuscript.

“Oxygen (O_2) plays a crucial role in the enzymatic reaction catalyzed by NOX and, therefore, could also contribute to the rate enhancement observed for Dbp1-NOX. Different microenvironments within the condensates may also affect O_2 saturation levels. However, given the poor water solubility of O_2 , it is unlikely that condensates are saturated with O_2 under the tested quiescent conditions.”

6. The kinetics of Dbp1-NOX in Fig 2A are clearly multi-phasic in contrast to the other proteins. This is the protein that the entire story hinges on, so I think we need an explanation for this. The initial fast burst phase is really what drives the difference between studies. Is there something fundamentally different about this reaction that make it multiphasic?

Answer from the authors: The curve for Dbp1-NOX follows a standard Michaelis-Menten behavior, and the initial fast burst is the expected faster substrate consumption at the beginning of the reaction, which slows down over time. We note that we considered the initial rates (when the substrate consumption is still linear) for all the different proteins, observing a faster reaction rate for the Dbp1-NOX system than for the other proteins.

Minor issues:

1. Could the sequences in the SI be added as text rather than images to make them machine-readable and easier to copy?

Answer from the authors: Sequences have been added in the SI as text.

2. 2. L. 171 – You cannot conclude that they scale exponentially from 3 data points.

Answer from the authors: We agree with the reviewer and have re-phrased in the new version of the manuscript.

“These values increase with the net charge of the fusion proteins (Figure 1F), confirming that electrostatic interactions are the main driving force for substrate partitioning.”

3. Fig 1F – Error bars missing

Answer from the authors: Thank you for the comment. Error bars have been added in the new version of the manuscript.

REVIEWERS' COMMENTS

Reviewer #1 (Remarks to the Author):

I am satisfied with the response of the authors and have no further concerns.

Reviewer #2 (Remarks to the Author):

The authors have addressed my comments and performed new analyses which strengthen the work, which was already impactful, including new NTA analysis, a phase diagram, information about the phase behavior of the protein, information about the enzyme surface charge, attempts to perform the nanoclusters in TEM, and additional controls. I appreciate the authors' effort in addressing my questions and suggestions and recommend on publication of the revised manuscript.

Reviewer #3 (Remarks to the Author):

The authors have done a nice job of addressing the reviewers' comments, and I support publication.